# Scalable room temperature incorporation of CO₂-selective ångström-scale pores in graphene for carbon capture

Ceren Kocaman[1], Luc Bondaz[1], Yueqing Shen[1], Ranadip Goswami[1], Mojtaba Chevalier[1], Jian Hao[1], Mounir Mensi[2] & Kumar Varoon Agrawal [1] ✉

Atom-thin porous graphene membranes offer unprecedented carbon capture performance thanks to Å-scale pores that combine ultrahigh permeance with attractive selectivity. However, incorporating a high pore density has until now required elevated-temperature ozone oxidation, while oxidation at room temperature was found to be sluggish, limiting scalability. Herein, we uncover that graphene oxidation by ozone is constrained by mass transfer of ozone and concentration polarization from the accumulation of reaction byproduct at the surface. We overcome this bottleneck using micro-channeled flow reactor that enhances mass transfer, accelerating the oxidation rate, leading to a tenfold higher pore density at room temperature. Centimeter-scale porous graphene with a high density of $CO_2$-selective pores is achieved, resulting in $CO_2/N_2$ selectivity up to 21 and $CO_2$ permeance up to 4050 gas permeation units. A brief subsequent room-temperature pore-expansion step further boosts performance. Our fully ambient, scalable protocol eliminates high-temperature equipment and provides a practical route to industrial production of porous graphene membranes for carbon capture.

Membrane-based point-source carbon capture is promising to cut the cost and energy consumption compared to conventional amine scrubbing-based carbon capture[1–6]. Among candidate materials, atom-thin porous graphene film hosting Å-scale pores are of interest[7–11]. It is an ideal selective layer for carbon capture membranes, thanks to the rapid permselective molecular transport from its zero-dimensional pores, allowing one to achieve attractive separation performances[7,12–16].

Several strategies have been developed to incorporate size-selective pores in graphene, including physical etching and chemical oxidation. Physical etching involving carbon knock out, incorporating vacancy defects in graphene[11,17–20]. Chemical etching by oxidative plasma[12,21–25], UV/O₃[26,27], O₂[28,29], KMnO₄/H₂SO₄[30], and O₃[31–35] has been demonstrated as a scalable approach to incorporate vacancy defects in graphene. Oxidation in O₃ has led to carbon-capture performance that compares favorably with other pore-formation methods and was

recently demonstrated for 50-cm²-sized membranes[36]. This is mainly because O₃ oxidation allows one to decouple the formation of pore precursors (epoxy clusters)[37] and pore formation by cluster gasification[38]. Ozone oxidation at elevated temperatures (e.g., at 80 °C) was needed to achieve an attractive combination of gas permeance and selectivity[36,39]. Conducting O₃ treatment at elevated temperatures requires heated reactors and stringent safety measures for hot, hazardous O₃, adding operational complexity that hinders scalable deployment. Room-temperature oxidation eliminates heating requirements, streamlines reactor design, enhances safety, lowers energy demand, and aligns the process with large-scale manufacturing.

The reaction of O₃ with graphene leading to pore formation occurs through a sequence of steps: chemisorption of epoxy as pore precursor, clustering of epoxy, and finally, pore opening at the center of the strained epoxy. The first step, epoxy chemisorption, involves a first-order reaction of O₃ with graphene[40]. The mobile epoxies,

[1]Laboratory of Advanced Separations (LAS), École Polytechnique Fédérale de Lausanne (EPFL), Sion, Switzerland. [2]Institute of Chemical Sciences and Engineering (ISIC), X-Ray Diffraction and Surface Analytics Platform (XRDSAP), EPFL, Sion, Switzerland. ✉e-mail: kumar.agrawal@epfl.ch

diffusing with an activation barrier of ~0.73 eV[41,42] on the graphene lattice, coalesce into organized epoxy trimer clusters to minimize the net energy of the oxidized graphene[37,43] As cluster size increases, the highly strained cluster core experiences C-C bond cleavage, generating ether linkages[39,44]. Subsequent energy input, e.g., light irradiation, gasifies the cluster core, resulting in pore opening at its centre[38,39]. Achieving a high $CO_2$ permeance requires a high density of pores, and therefore, a high density of epoxy clusters, and by extension, a high density of epoxy. Therefore, ensuring a high chemisorption rate of epoxy is crucial.

However, the reaction for pore precursor nucleation was found to be markedly sluggish at room temperature[31,33]. The elementary first-order reaction of ozone with graphene proceeds with an estimated rate constant of ~1.4 s$^{-1}$ and an activation energy of 0.66 eV[38,40]. During this reaction, the accumulation of $O_2$ byproducts near the surface creates a concentration polarization layer, limiting the local $O_3$ concentration and slowing down oxidation kinetics. This highlights the need for mass transfer–enhancing strategies to achieve effective oxidation at ambient conditions.

Herein, motivated by developing a scalable pore nucleation protocol, we probed the role of ozone mass transfer to graphene in controlling the oxidation reaction. We identify the underlying bottleneck; reaction-generated $O_2$ accumulates at the graphene surface, creating concentration polarization that limits $O_3$ mass transfer. We overcome this limitation by developing and implementing a simple confined reaction environment using micro-channeled slits which accelerate gas velocity and mass transfer by severalfold. A brief, room-temperature second cycle ozone exposure step results in pore expansion which further enhances membrane performance.

Unlike prior approaches that rely on elevated temperatures (~80–90 °C)[36] for ozone-based oxidation, we achieve Å-scale pore incorporation entirely at room temperature. This is accomplished without increasing reaction time, achieving oxidation in 1 h. The resulting centimeter-scale membranes exhibit attractive $CO_2/N_2$ separation performances. By eliminating the need for high-temperature processes and enabling efficient pore formation at room temperature, this approach highlights a scalable and practical pathway for preparing porous graphene membranes suitable for gas separation applications.

## Results

### Ozone treatment of graphene at room temperature

Literature on pore density in graphene as a function of oxidation temperature shows that density increases as a function of temperature, and highlights the role of rapid chemisorption kinetics at higher temperatures[38]. For example, a $CO_2$-permeable pore density of $4 \times 10^{11}$ cm$^{-2}$ could be obtained using $O_2/O_3$ gas mixture as the oxidizing agent at a temperature of 80 °C[39]. As we detail below, oxidation using ozone at room temperature yields negligible pore density under conventional conditions.

Ozone oxidation rate is a function of temperature-dependent first-order kinetics, with a rate constant of ~1.4 s$^{-1}$ at room temperature[40]. It is also dependent on the rate of ozone molecules impinging on the surface, which is determined by the local $O_3$ concentration[40,41]. Therefore, to achieve a high density of pores on the graphene lattice at room temperature, increasing local $O_3$ concentration by increasing mass transfer is an effective strategy. For this, slit reactors hosting sub-millimeter-sized gaps were designed. Two different reactor geometries were used, with slit gaps of 1000 or and 400 μm. Chemical vapor deposition (CVD) derived single-layer graphene, resting on Cu foil, with size of $2 \times 8$ cm$^2$ was placed inside the slit reactor. The top side of the slit designed with two sides legs that secure samples with a width of 2 cm when assembled on top. These legs not only hold the samples in place but also provide the defined flow gap (Figure S1). Considering the thickness of Cu foil (~100 μm), this resulted in a flow channel (FC) gap of 900 and 300 μm, respectively. Accordingly, these slits are termed as FC900 and FC300, respectively. Samples smaller than 2 cm size also remained fixed inside the slit at varying flow rates, due to high frictional resistance (Supplementary Note S1).

The slit was placed inside a cylindrical 1-inch diameter tube connecting to a supply of $O_3$, resulting in generation of O clusters. Pores were created by gasification of the cluster in 390 nm (3.2 eV) light[39]. A schematic of the oxidation of the graphene lattice followed by photonic gasification at room temperature is shown in Fig. 1.

During gas flow over graphene, a diffusion boundary layer evolves at the surface that governs the $O_3$ flux from the bulk stream to the surface. Given that $O_3$ feed is a dilute mixture of $O_3$ in $O_2$ (9.4 mole%, limited by the productivity of the $O_3$ generator), and each reaction event releases an additional $O_2$ molecule as a byproduct[40], the near-surface $O_3$ concentration is markedly lower than the bulk stream value, making the reaction diffusion-controlled. The Reynolds number (Re) for the flow is low (~7-11), resulting in a fully developed flow in the laminar regime (Supplementary Note S2, Table S1). In this regime, the mass transfer coefficient, and hence the $O_3$ flux to graphene surface, scales with velocity (v) as $v^{1/3}$. A higher v both increases flux and thins the concentration boundary layer[45,46]. Therefore, increasing v is an

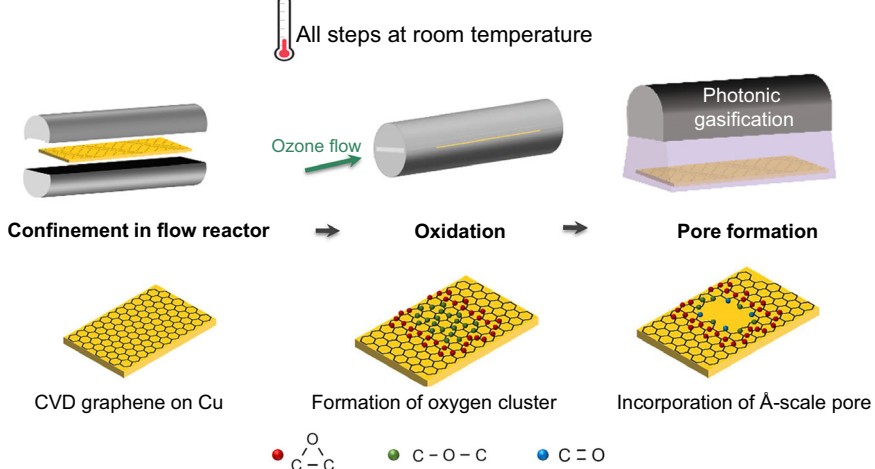

**Fig. 1 | Schematic of ozone oxidation inside mircrofluidic reactor for pore incorporation at room temperature.** Graphene is exposed to $O_3$ under confined flow to promote the formation of epoxy clusters through surface oxidation. These clusters are then gasified using 390 nm light resulting in Å-scale pores. The entire process is conducted at room temperature, enabling scalable and ambient fabrication of porous graphene membranes.

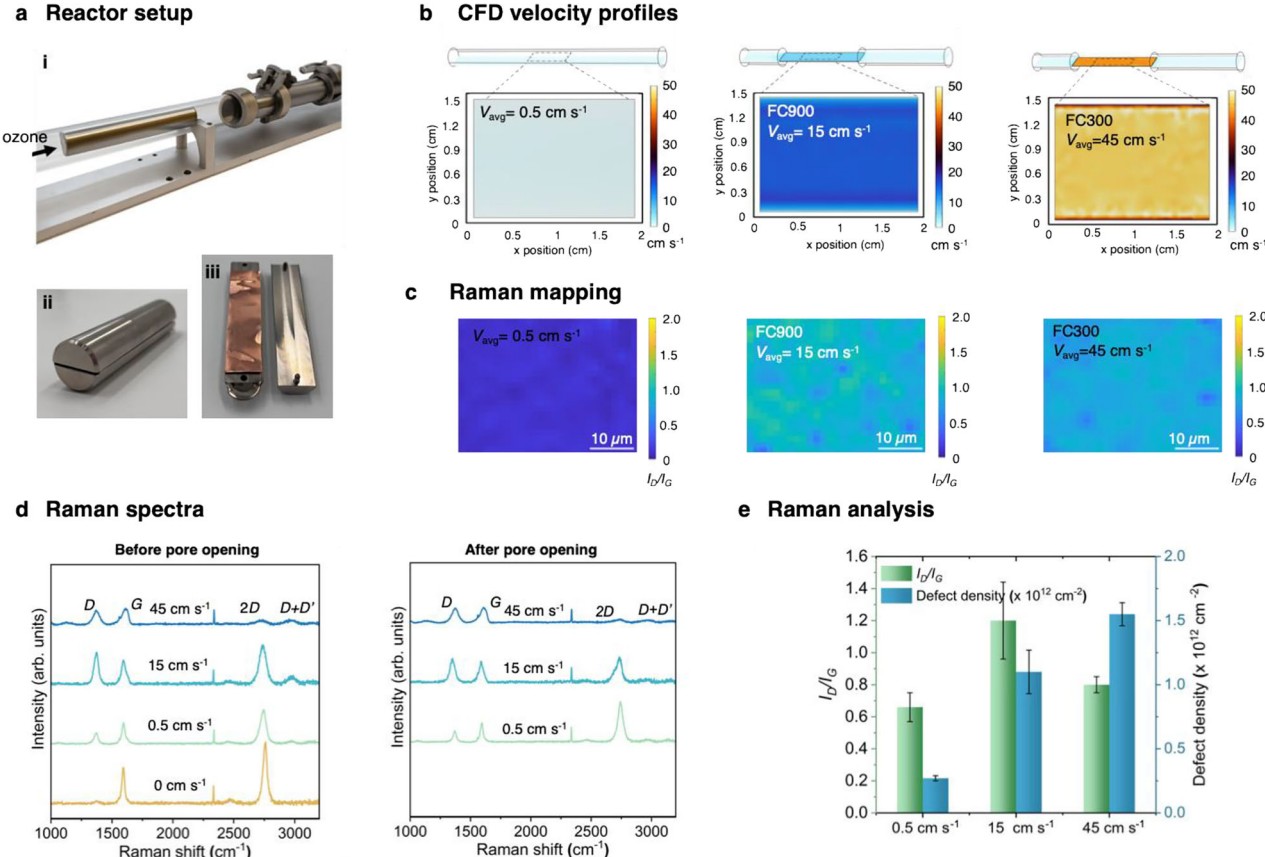

**Fig. 2 | Effect of confined ozone flow on room-temperature oxidation of graphene. a** Schematic of experimental setup (i) and images of custom flow channel with a 900 μm gap (FC900) used to confine $O_3$ flow over the graphene surface (ii-iii). **b** COMSOL simulations 2D velocity profiles at 100 μm above the graphene for three conditions: no flow channel, FC900 (900 μm gap), and FC300 (300 μm gap). **c** Raman spectroscopy mapping of $I_D/I_G$ for samples exposed to ozone flow under different conditions: without a flow channel, with FC900, and with FC300

**d** Representative Raman spectra before and after photonic gasification for the three velocities, showing evolution of *D*, *G*, and *2D* peaks. **e** Quantification of evolution of the $I_D/I_G$ ratio and defect density, analyzed based on the carbon amorphization trajectory, for the three velocities. Error bars represent the standard deviation from the average of twenty-five independent spectral points for each sample. Source data are provided as a Source Data file.

attractive avenue for increasing flux[47]. To exploit this dependence, we confined the gas in slit reactors with 900 μm and 300 μm FCs (Fig. 2a).

Computational fluid dynamics (CFD) simulations of $O_3/O_2$ mixture flow were performed using COMSOL to understand the effect of using FC on the flow dynamics. The 3D models and enlarged 2D velocity profiles at a height of 100 μm above graphene are shown in Fig. 2b, demonstrating a uniform $v$ profile across the samples. The profiles show that FCs drastically increases the average $v$ near the surface, rising from 0.5 cm s$^{-1}$ (without a flow channel) to 15 cm s$^{-1}$ (with FC900) and further increasing to 45 cm s$^{-1}$ (with FC300). Correspondingly, the boundary layer thickness decreases from ~4.3 mm (without a flow channel) to ~0.12 mm (with FC300). This transition is captured by the Péclet number, which increases from ~5 to ~300, indicating a shift from diffusion-limited to convection-enhanced transport. Such enhancement is expected to improve the mass transfer of ozone to the graphene surface. (Supplementary Note S2, Table S1).

To evaluate the effect of this improved transport on oxidation kinetics, we performed micro-Raman spectroscopy mapping over a 40 μm × 30 μm area. Functionalization with epoxy groups results in sp$^3$-hybridized bonds, which activate the breathing mode of the six-atom ring in graphene, resulting in the *D* peak in Raman spectra[48]. Indeed, *D* peak intensity increased after ozone exposure. The concurrent decrease in $I_{2D}/I_G$ peak intensity is consistent with the formation of sp$^3$-hybridized sites (Fig. 2c)[48–50]. With increasing $v$, the changes in the

Raman peaks were pronounced, resulting in broadening of *D* and *G* peaks and a decrease in the *2D* peak intensity. The $I_D/I_G$ ratio increased from 0.7 ± 0.1 ($v$ of 0.5 cm s$^{-1}$) to 1.2 ± 0.24 (15 cm s$^{-1}$) and then decreased to 0.8 ± 0.05 (45 cm s$^{-1}$). This initial increase, followed by a decrease, is attributed to an increase in the defect (sp$^3$-hybridized bonds) density where the structure transitions from the graphitic regime to the nanocrystalline regime[50]. Indeed, the 2D peak for the sample oxidized using the highest $v$ plummets. $I_D/I_G$ maps indicated that the oxidation was uniform over the entire area (Fig. 2c). These characterizations reveal that increasing $v$ is an effective strategy for oxidation at room temperature.

To convert the oxidized clusters into pores, we irradiated the ozone-treated graphene for 5 s with 390 ± 20 nm light at 1.7 W cm$^{-2}$, using a custom lamp that uniformly illuminated a 10 cm×10 cm area (Figure S3). The photons selectively gasify the ether groups at each O-cluster core, creating a core shell structure with pore at the core of the structure[39]. Given that this mainly gasifies ether groups, the Raman spectra after the photonic gasification did not change drastically (Fig. 2d). Based on the carbon amorphization trajectory, the density of defects in the gasified samples monotonically increased from 0.3 × 10$^{12}$ cm$^{-2}$ ($v = 0.5$ cm s$^{-1}$) to 1.1× 10$^{12}$ cm$^{-2}$ ($v = 15$ cm s$^{-1}$) and further to 1.6 × 10$^{12}$ cm$^{-2}$ ($v = 45$ cm s$^{-1}$, Fig. 2e, Supplementary Note S3). Importantly, the defect densities before and after photonic gasification were nearly identical (Figure S4), indicating that the increase in defect density arises primarily from the oxidation step, rather than from the

lattice gasification process. These results confirm that the slit-reactor geometry effectively increases oxidation and the associated defect density in the gasified samples.

To investigate how ozone exposure at room temperature functionalizes graphene, we performed X-ray photoelectron spectroscopy (XPS). To minimize the atmospheric contamination on the samples, the samples were transferred to a custom XPS sample holder in less than one minute and were maintained under vacuum until introduction into the XPS chamber. The wide spectra for each condition are shown in Figure S5. The absence of a characteristic O 1s peak in the 528–538 eV range for pristine graphene confirmed that the samples were transferred with minimized contaminations (Figure S6). C1s peak of $O_3$-treated graphene samples showed an asymmetric peak, referenced at 284.2 eV with a shoulder corresponding to epoxy/ether functional groups, consistent with $O_3$-driven oxidation (Fig. 3)[31,51]. Based on the $V^{1/3}$ scaling of the mass transfer coefficient, we estimate that the use of confined microchannel flow increases the ozone mass transfer coefficient by a factor of ~3.1 (FC900) and ~4.5 (FC300) compared to the unconfined condition. Correspondingly, the intensity of C-O peak increased with increasing mass transfer (Fig. 3b-d), with the highest population of C-O ( ~13%) with the highest $v$ (45 cm s$^{-1}$), consistent with the Raman data, confirming a significant enhancement in mass transport under these flow configurations (Supplementary Note S4, Table S2). A minor carbonyl component was also present, arising from unavoidable air exposure during handling. Notably, this carbonyl signal remains constant across all conditions, whereas C−O grows with enhanced $O_3$ delivery, indicating the important role of mass transfer in functional-group formation.

## Evaluation of pore density

Direct visualization of the oxidized graphene lattice is a powerful tool for verifying whether the mass-transfer-enhanced protocol generates the Å-scale $CO_2$-selective pores. For this, we used aberration-corrected high-resolution transmission electron microscopy (AC-HRTEM) to image the resulting graphene lattice. Imaging was performed using an 80 keV electron beam with an electron dose of 0.9–1.3 nA to eliminate the possibility of pore formation and expansion during imaging. A survey of images confirmed an increase in density of vacancy defects from $0.4 \times 10^{12}$ cm$^{-2}$ ($v = 0.5$ cm s$^{-1}$) to $3.2 \times 10^{12}$ cm$^2$ ($v = 15$ cm s$^{-1}$) and $4.5 \times 10^{12}$ cm$^{-2}$ ($v = 45$ cm s$^{-1}$), consistent with the increasing density trend from Raman and XPS. Examples of AC-HRTEM images are shown for each condition in Figure S8-S10. The vacancy density in these room-temperature oxidation samples exceeds that reported for oxidation at elevated temperatures (Fig. 4a)[38], independent of whether thermal or photonic gasification was used. For instance, photonic gasification after 43 °C oxidation yielded a lower density of $2.5 \times 10^{12}$ cm$^{-2}$[39]. The high pore densities achieved at room temperature underscore the important role of mass transfer.

The vacancy defects comprise those that are permeable to $CO_2$ as well as those that are too small to be permeable. To evaluate the density of $CO_2$-permeable pores, defects were classified by the number of missing carbon atoms, e.g., a defect missing ten carbon atoms was labeled pore-10. Based on literature on $CO_2$-transport from functionalized graphene pores[52], only pores with size equal to or larger than pore-10 were considered $CO_2$-permeable; smaller pores were classified as impermeable due to insufficient electron density gap for $CO_2$ permeation. Based on this, examples of $CO_2$-permeable and impermeable pores are shown in Fig. 4b where red outlines mark the estimated pore edges, while gray dots denote the absent carbon atoms.

In graphene oxidized without flow confinement ($v = 0.5$ cm s$^{-1}$), most defects were monovacancies (Fig. 4c), impermeable to $CO_2$, accounting for the low gas permeance observed despite an overall 0.5 of $0.4 \times 10^{12}$ cm$^{-2}$. However, pore density increased significantly with increasing average velocity, as indicated by the increase in the relative

mass transfer coefficient, which scales as $k_m \propto V^{1/3}$. (Supplementary Note S4, Figure S11) Higher $v$ (15 cm s$^{-1}$) under FC900 produces a markedly higher density of $CO_2$-permeable pores (1.4 $\times 10^{11}$ cm$^{-2}$; Fig. 4c). Further increasing $v$ to 45 cm s$^{-1}$ in FC300 boosts this population to $2.2 \times 10^{11}$ cm$^{-2}$. Thus, confined-flow oxidation shifts the defect population toward permeable pores, crucial for achieving attractive separation performance.

## Controlled pore expansion at room temperature

Room-temperature oxidation inside the flow channels produces both gas-permeable and impermeable vacancies, as evident in AC-HRTEM images. Although confining the $O_3$ flow raises the density of $CO_2$-permeable pores from negligible to $2.2 \times 10^{11}$ cm$^{-2}$ (FC300), impermeable defects still dominate, restricting membrane flux. Thus, a selective enlargement of the pre-existing, nonpermeable vacancies is attractive to boost separation performance.

We therefore applied a second $O_3$ cycle at room temperature, exploiting the reactivity differences for C at and away from pore edges. At room temperature, the energy barrier (1.1-1.3 eV) for creating new pores inside epoxy clusters is too high to overcome. In contrast, existing pores, resting on Cu foil, are significantly more reactive toward $O_3$. This is because when graphene rests on a Cu substrate, electron puddles formed by charge transfer locally dope the graphene. This doping lowers the energy barrier for gasification near the pore edges upon $O_3$ exposure. This creates a favorable condition for the selective expansion of existing pores[34,53,54].

Specifically, graphene first underwent 1 h $O_3$ exposure in FC900 at 25 °C, followed by 5 s photonic gasification to form porous graphene, before a second ozone treatment in the same reactor for varying times (5–30 min, Fig. 5a).

Even 5 min of post-ozone exposure nearly doubled the $I_D/I_G$ ratio from 1.10 to 2.25. Longer treatments (15–2 h) further broadened the D, G, and 2D bands, diminished 2D peak intensity, and produced a pronounced ($D+D'$) band, signifying greater lattice disorder (Fig. 5b, Figure S12).

Carbon amorphization trajectory[50] analysis shows that the estimated average defect distance ($L_d$) shrank steadily from 8.7 to 5 nm as the 2nd cycle exposure time increased from 0 to 30 minutes (Fig. 5b, Supplementary Note S3). This progressive decrease in $L_d$ reflects an increase in the overall porosity through pore enlargement. AC-HRTEM examination after a 15 min $O_3$ treatment clarifies this evolution (Fig. 5c). The total defect density is comparable to that from the sample that did not see the 15 min $O_3$ treatment ($3.2 \times 10^{12}$ vs $3.5 \times 10^{12}$ cm$^{-2}$), yet the $CO_2$-permeable pore density increases, from $1.4 \times 10^{11}$ to $2.2 \times 10^{11}$ cm$^{-2}$ (Fig. 5e, Figure S13). This confirms that the second $O_3$ cycle mainly expands existing vacancies. This controlled expansion strategy is particularly beneficial for enhancing gas separation performance (discussed in the next section), as it increases the number of $CO_2$-permeable pores without compromising the integrity of the graphene lattice with excessive new defect formation (Fig. 5d-e).

## $CO_2/N_2$ separation

For the preparation of graphene membranes, large-area graphene coupons ($2 \times 8$ cm$^2$) were oxidized at 25 °C, then photonic gasified to incorporate pores. Optionally, the 2nd cycle of $O_3$ treatment at 25 °C was carried out to increase the density of permeable pores. A poly(-trimethylsilylpropyne) (PTMSP) based mechanically reinforcing film (MRF, 1 μm thick) was deposited on porous graphene, after which a thermal-release tape (TRT) was laminated on top of the PTMSP to provide additional mechanical support and facilitate easy handling. The TRT/PTMSP/graphene/Cu layer was then electrochemically delaminated from the Cu and transferred onto a porous polyethersulfone (PES) support. Finally, TRT was removed by heating the TRT/PTMSP/graphene/PES stack at 130 °C for 2 min, resulting in PTMSP/graphene/

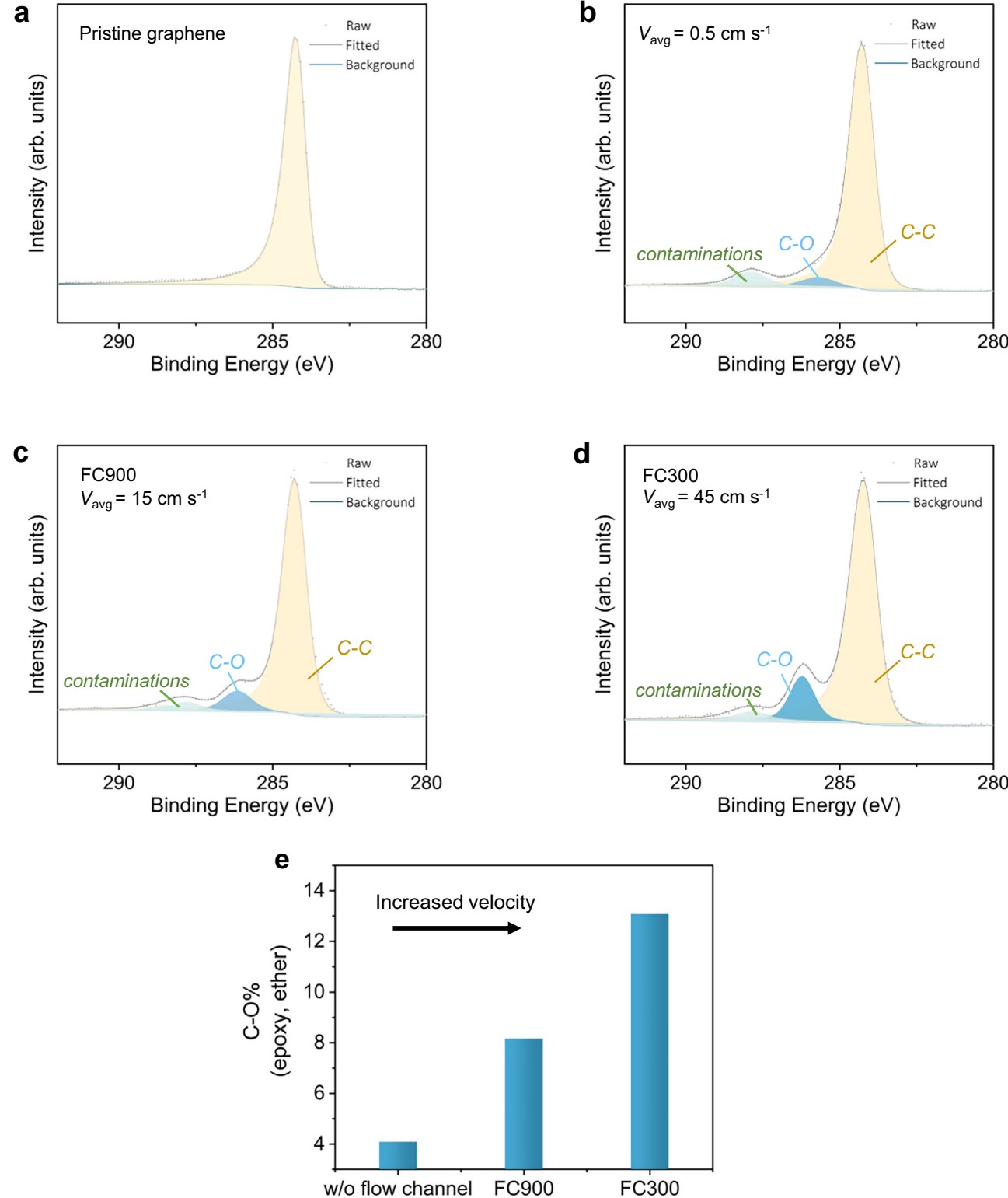

**Fig. 3 | Oxygen functional group evolution on graphene samples after ozone treatment under various conditions at room temperature. a** XPS C 1 s spectrum of pristine graphene. **b**–**d** C 1 s spectra of graphene samples exposed to ozone at room temperature: **b** without a flow channel, **c** with the FC900 flow channel, and (**d**) with the FC300 flow channel. **e** Relative C−O content as a function of ozone flow conditions, illustrating the impact of flow channel configuration (velocity) on the degree of graphene oxidation. Source data are provided as a Source Data file.

PES membranes. A picture of ~4.5 cm × 1.5 cm sized membrane coupon is shown in Fig. 6a.

To compare oxidation protocols, centimeter-scale coupons were mounted in a 1-cm-diameter membrane module (Fig. 6b). Importantly, each oxidation batch yielded at least three replicate samples of consistent quality, confirming the scalability of the process to larger areas and enabling batch-level reproducibility.

The separation performance of porous graphene was assessed using a gas transport resistance model (Supplementary Note S5, Table S3), allowing precise estimation of the permeance of the porous

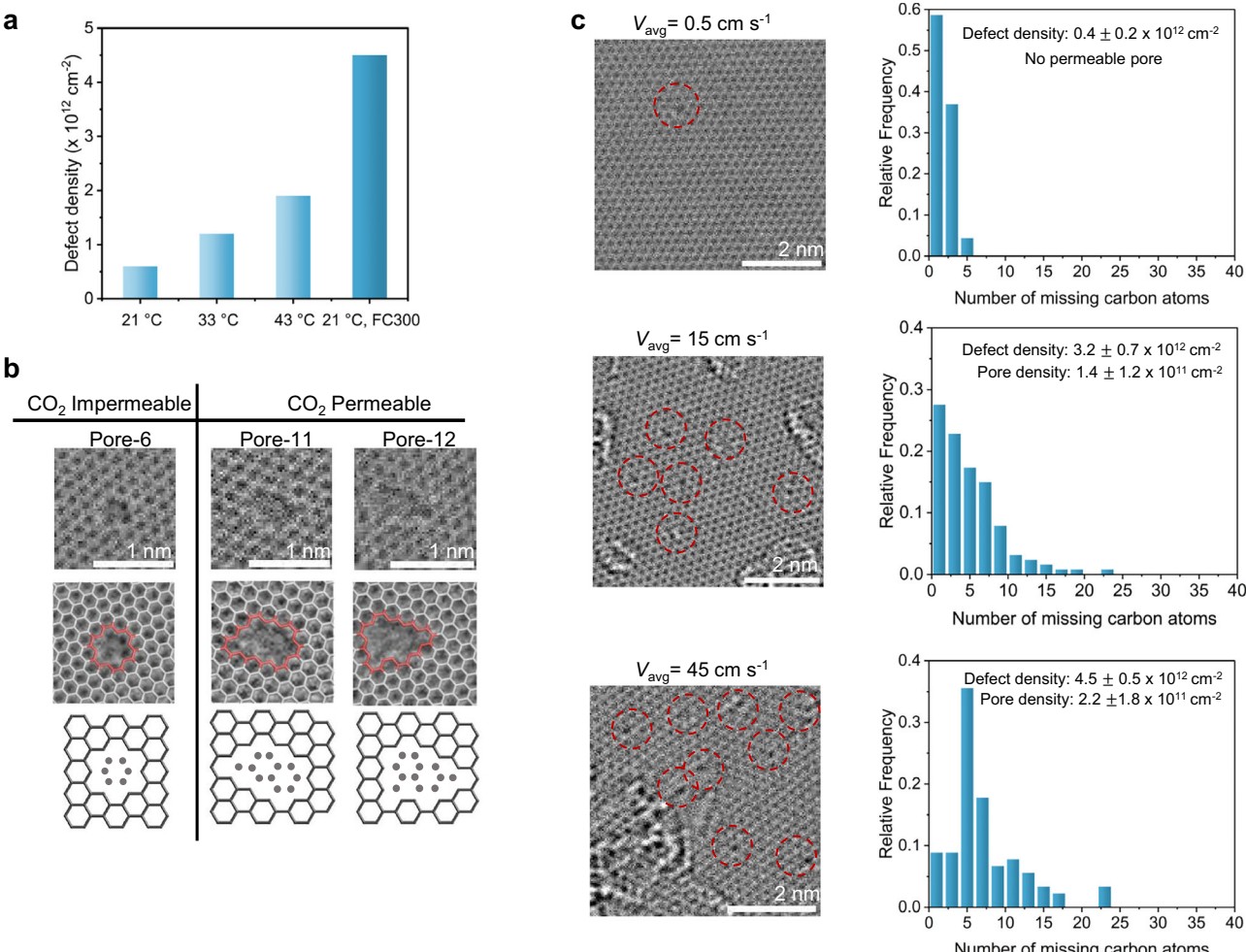

**Fig. 4 | Structural analysis of graphene pores formed by room temperature oxidation under different flow conditions. a** Comparison of the oxidative etching approach used in this study with literature. **b** Examples of $CO_2$-permeable and impermeable pores formed after 1 h oxidation at room temperature using FC900. Middle row outlines the estimated pore geometry, and the bottom row illustrates atomic-scale pore structures with missing carbon atoms marked in gray. **c** AC- HRTEM images of graphene oxidized for 1 h under three conditions: without a flow channel, with FC900, and with FC300. Red circles indicate defects on the graphene lattice. Right panel shows pore size distribution based on the number of missing carbon atoms, quantified from AC-HRTEM images for each oxidation condition. Source data are provided as a Source Data file.

graphene[36,55]. Membranes fabricated without FC ($v = 0.5$ cm s$^{-1}$) exhibited low permeances with three membranes yielding an average $CO_2$ permeances of $64 \pm 11$ GPU (1 GPU = $3.35 \times 10^{10}$ mol m$^{-2}$s$^{-1}$Pa$^{-1}$), consistent with transport solely from intrinsic vacancy defects in graphene. In contrast, higher $v$ in FC900 and FC300 significantly enhanced $CO_2$ permeance of porous graphene to above 1000 GPU (Fig. 6c). These trends align with observations from AC-HRTEM, which showed a higher density of $CO_2$-permeable pores with improved mass transfer. In addition to enhanced permeance, the $CO_2/N_2$ selectivity of porous graphene also improved. Membranes prepared with high $v$ oxidation yielded selectivities of $17.8 \pm 1.1$ and $18.5 \pm 4$, respectively. When the oxidation duration was increased from 1 to 2 h using FC900, $CO_2$ permeance further increased above 2000 GPU (Figure S14). However, this was accompanied by a decrease in $CO_2/N_2$ selectivity, indicating the formation of larger pores likely due to coalescence of nearby pores. Membranes exhibited a modest decline in $CO_2$ permeance (~15%) over 60 h of testing. However, a simple thermal treatment at 130 °C for 2 h enabled partial performance recovery (Figure S15).

The 2$^{nd}$ cycle of $O_3$ oxidation further enhanced performance (Fig. 6d). FC900 films exposed to $O_3$ for 5 min and 15 min resulted in a marked improvement in gas separation performances. $CO_2$ permeance of porous graphene increased significantly to $3646 \pm 364$ GPU and $3800 \pm 142$ GPU, after 5 and 15 min of exposure, respectively. Correspondingly, the $CO_2/N_2$ selectivity also improved, to $17.3 \pm 1.6$ and $21.1 \pm 1.1$, respectively.

Graphene membrane prepared by FC900 followed by 2$^{nd}$ cycle of $O_3$ oxidation also tested with gas mixture stream (50% $CO_2$:$N_2$) and humidified feed stream When 3% water vapor was present in the feed, $CO_2$ permeance dropped by ~35% relative to dry conditions, but interestingly $CO_2/N_2$ selectivity increased by ~50% (Figure S16 The combination of high $CO_2$ permeance and favorable $CO_2/N_2$ selectivity achieved here indicates the potential of single-layer graphene membranes, and highlights the importance of scalable fabrication methods to enable their industrial adoption (Figure S17, Table S4).

The improvements are also closely aligned with the structural transformations observed via AC-HRTEM. As discussed, the 2$^{nd}$ $O_3$ treatment cycle does not generate a significant number of new defects, but instead promotes the enlargement of pre-existing vacancies into gas-permeable pores. The corresponding enhancement in $CO_2/N_2$ selectivity from 17.8 to 21.1 further confirms that the enlarged pores

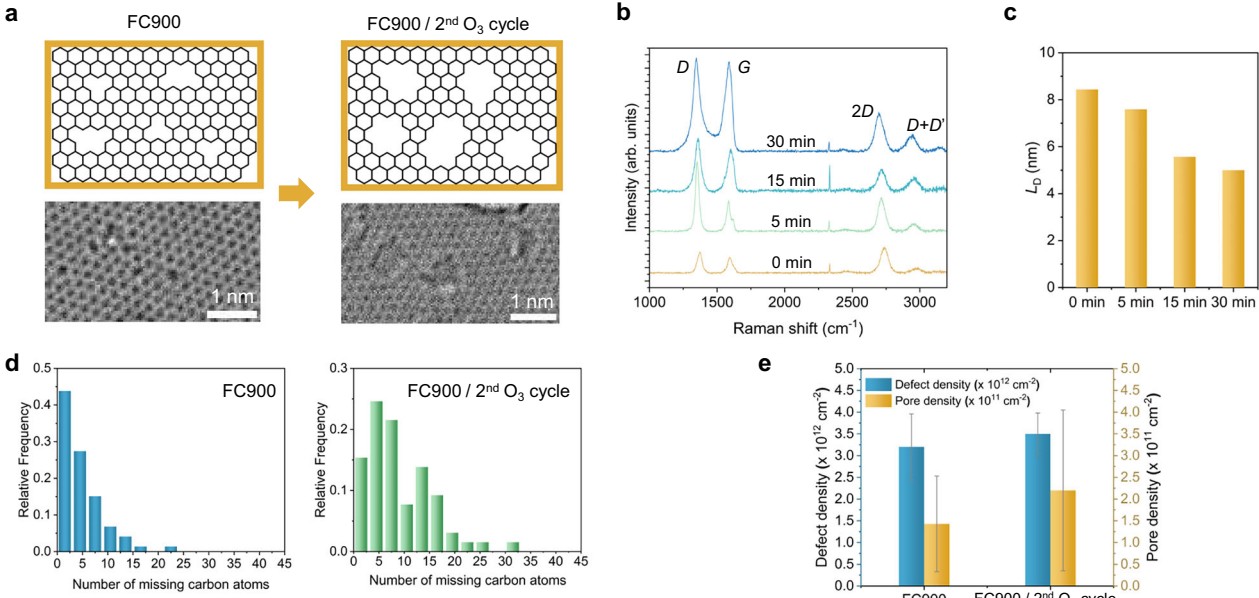

**Fig. 5 | Controlled pore expansion on graphene samples subjected to sequential ozone oxidation, after photonic gasification. a** AC-HRTEM images of graphene samples after initial oxidation with FC900, and 15-minute post-treatment in ozone (FC900/ 2nd O3 cycle). **b** Raman spectra of graphene samples treated with ozone using FC900 for 1 h, followed by photonic gasification and post-ozone exposure for varying durations (0, 5, 15, and 30 minutes). **c** Estimated inter-defect distance ($L_D$) after the second ozone exposure cycle, calculated using the carbon amorphization trajectory. **d** Pore size distribution of graphene samples treated under the FC900 and 15-minute post-treatment in ozone (FC900/ 2nd O3 cycle), estimated from AC-HRTEM images. **e** Defect and pore density of graphene samples oxidized with FC900 and 15-minute post-treatment in ozone (FC900/ 2nd O3 cycle). Error bars represent the standard deviation of the average defect density and pore density for each sample, calculated from a scanned HRTEM area of 8820 nm². Source data are provided as a Source Data file.

remain within the optimal size range for $CO_2/N_2$ separation. This is further established by the fact that when the 2nd cycle time was extended to 30 min, $CO_2/N_2$ selectivity declined to 7, from over-enlargement of pores. This highlights the importance of precisely tuning the duration of post-$O_3$ exposure to balance pore density with selectivity.

## Discussion

We demonstrate a fully ambient, scalable route for generating $CO_2$-selective pores in single-layer graphene by coupling ozone oxidation with confined-flow mass-transfer and a brief photonic gasification step. By simply narrowing the flow gap to intensify local gas velocity, we overcame concentration-polarization of $O_3$ at the graphene interface. We increased the oxidation rate at room temperature by approximately an order of magnitude under our conditions. Furthermore, the controlled expansion of existing non-selective pores allowed convenient tuning of pore size at room temperature. The centimeter-scale membranes demonstrated here indicate the method's uniformity and scalability. This method reduces the complexity of the pore incorporation protocol in graphene, providing a straightforward and scalable pathway to porous graphene membranes for carbon capture. More broadly, the ability to tune pore density and size in ambient conditions expands the potential use of nanoporous atom-thin membranes in gas separation.

## Methods

### Pore Incorporation on the Graphene Lattice

Single-layer graphene samples, prepared by atmospheric pressure CVD on ~100 μm thick Cu foil, were purchased from General Graphene. Before the ozone treatment, as-received graphene samples were annealed in the presence of $H_2$ at 1 bar maintained by 50 sccm flow for 1 hours at 800 °C, to remove surface impurities and to reduce copper oxide. Subsequently, 2 × 8 cm² graphene samples were introduced inside the FCs. The images of the flow channels are shown in Fig. 2a. These flow channels were inserted into a tubular quartz reactor maintained at ambient pressure (~1 bar) and connected to an ozone generator (Absolute Ozone, Atlas 60). For oxidation, samples were exposed to of 100 sccm of a 9.35 mol% $O_3/O_2$ gas mixture for 1 hour at room temperature (no additional sweep gas was used). No pressure gradient was applied across the graphene during this process. Following the $O_3$ treatment, for lattice gasification, the oxidized samples were exposed to light (390 nm, 3.2 eV) for 5 s. For samples undergoing post-$O_3$ treatment, after photonic gasification, the porous graphene samples were exposed again to $O_3$ flow in the same flow channels for a specified duration.

### Preparation of Graphene Membranes

Graphene/Cu samples were coated with 3 wt% PTMSP (abcr GmbH) in toluene, which is a high free-volume polymer that act as the mechanical support for graphene. The PTMSP/graphene samples were then transferred to the target commercial porous polymeric support (PES, with 0.2–0.8 μm-sized pores) using TRT, after electrochemical delamination from the Cu support. After drying, the TRT was removed by simply heating the samples to 130 °C[56].

### Gas Permeation Tests

Graphene membranes with 2 x 8 cm² in size (the size that can fit into the flow channels but can be scalable to desired size) were tested in centimeter-scale membrane modules. Membranes were sandwiched between custom-made stainless-steel modules with 1 cm diameter.

Membranes were tested in custom-made single component permeation setups with pure $CO_2$ and $N_2$ at 2 bar feed pressure. For each test, the membrane module was heated in presence of $CO_2$ to 130 °C and kept there for 1 h to clean possible atmospheric contaminants on the membrane surface. After heating, the membrane was cooled down to the room temperature for testing.

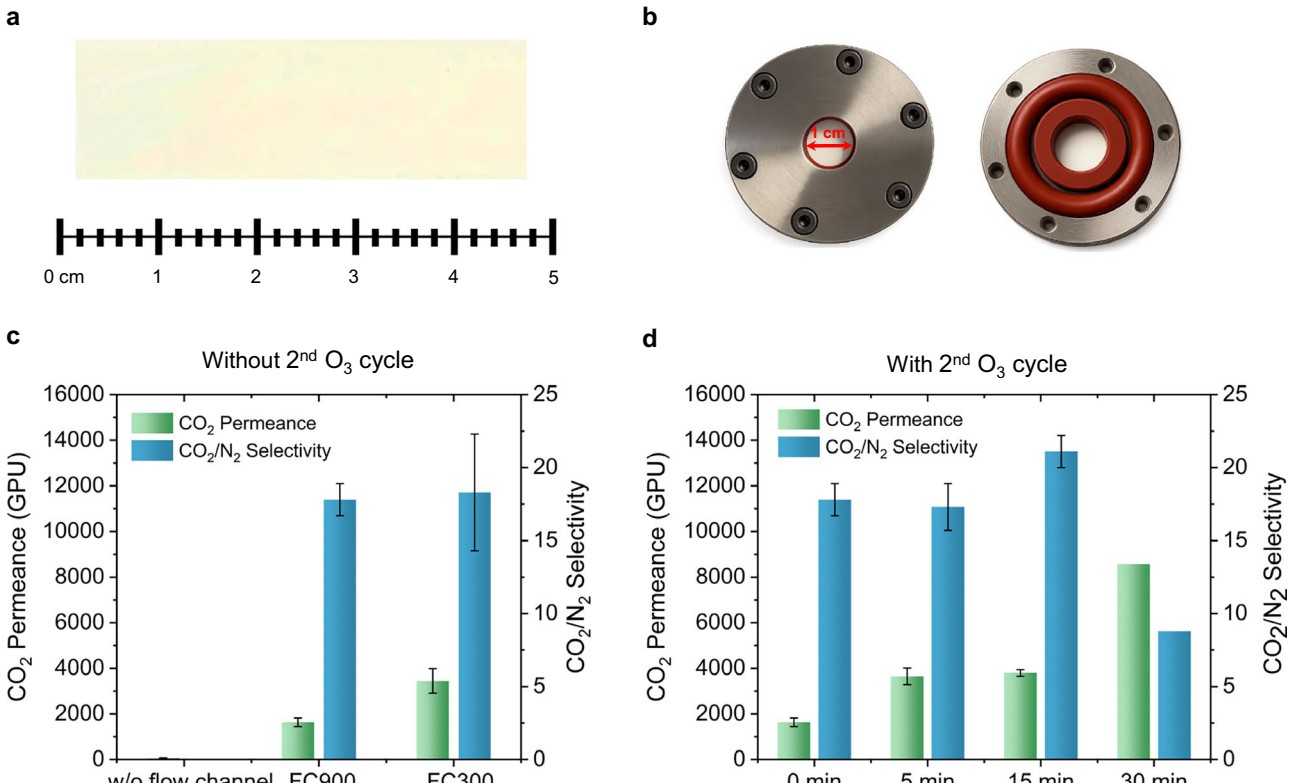

**Fig. 6 | Performance evaluation of centimeter-scale porous graphene prepared under various oxidation conditions. a** Picture of ~4.5 cm × 1.5 cm sized membrane coupon prepared by the room temperature oxidation protocol. **b** Architecture of membrane module with 1 cm diameter. **c** Gas separation performance of porous graphene fabricated from samples oxidized under different flow conditions: without a flow channel, with FC900, and with FC300. The error bars represent the standard deviation of the permeance and selectivity across at least three graphene membranes. **d** Separation performance of porous graphene produced via controlled pore expansion. Samples were initially oxidized using FC900, followed by lattice gasification and subsequent pore enlargement at room temperature using FC900 for varying durations. The error bars refer to the standard deviation in the permeance and selectivity of at least three membranes. The separation performance of porous graphene was assessed using a gas transport resistance model (Supplementary Note S4, Table S3). Source data are provided as a Source Data file.

## CFD Simulations

The gas flow inside confined flow channels were simulated using COMSOL Multiphysics 6.1. For the simulations, $O_2$ was selected as the fluid material. It was assumed that there is no slip boundary condition on the flow channel walls, flow is in the laminar flow region and the gas discharges to the atmospheric pressure.

## Characterizations

Raman spectroscopy performed using Renishaw inVia™ spectroscope with a blue laser ($\lambda = 457$ nm) and 100x objective. Curve fitting and peak intensity calculation were performed with MATLAB.

X-ray photoelectron microscopy performed using a Mg Kα X-ray source (1253.6 eV) and Phoibos 100 (SPECS) hemispherical electron analyzer with multichanneltron detector. To minimize air adsorption and atmospheric purities on graphene samples, right after the $O_3$ treatment, samples transferred in a glove box to custom-made XPS sample holder that can be kept under vacuum. The XPS spectra processed using CasaXPS, for background subtraction Shirley method was used.

For visualization of graphene lattice and pores, Titan Themis AC-HRTEM with a Wein-type monochromator was utilized. To avoid expansion of the existing pores, 80 keV incident electron beam was used during imaging. Sample preparation for imaging was explained in our previous work[33,57]. Before imaging, TEM grids were cleaned in $H_2$ environment at 600 °C for 1 h. The incident beam was monochromated to minimize the chromatic aberration. The defect density and pore size of defects were estimated with the graphical method we previously developed[34], in which the number of missing carbon atoms is determined from the vacancy area in AC-HRTEM images using the atomic density of pristine graphene. For defect density and pore size distribution analysis, a total area of approximately 25200–31500 nm² per sample was analyzed from high-resolution AC-HRTEM images. For PSD calculations, only pores not intersecting contamination were included to ensure accurate size estimation.

## Data availability

The data supporting the findings of this study are included in the paper and its supplementary information files. Source data are provided with this paper.

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

## Acknowledgements

We acknowledge the host institute, École Polytechnique Fédérale de Lausanne (EPFL) for generous support. K.V.A. acknowledges EPFL Solutions4Sustainability grant on developing scalable fabrication protocol for graphene membranes. K.V.A. also acknowledges Swiss National Science Foundation Assistant Professor Energy Grant (PYAPP2_173645) for supporting parts of this part. We thank the EPFL mechanical workshop for fabricating the tools needed in this project.

## Author contributions

K.V.A. and C.K. conceived the project and wrote the manuscript. C.K. prepared membranes, conducted Raman and XPS measurements, performed CFD simulations, and carried out image and data analysis. L.B. conceived and designed the customized large-area light source. L.B. and Y.S. carried out AC-HRTEM imaging. R.G. assisted in membrane mixed gas separation measurement. M.C. provided input for Raman data analysis. J.H. assisted in the interpretation of CFD simulation results. M.M. assisted with XPS sample preparation and data interpretation. All authors discussed the results and provided feedback on the manuscript.

## Competing interests

K.V.A. and M.C. are cofounders of spinoff looking to commercialize porous graphene membrane based carbon capture. The remaining authors declare no competing interests.
