## [Transparent Peer Review file · Nature Communications]

Scalable room temperature incorporation of CO₂-selective ångström-scale pores in graphene for carbon capture

Corresponding Author: Professor Kumar Varoon Agrawal

Version 0:

Reviewer comments:

Reviewer #1

(Remarks to the Author)

Nil

(see attached)

Reviewer #2

(Remarks to the Author)

The study addresses a mass transfer-limited challenge, rather specific to the authors' system for O₃-induced nanopore formation in monolayer graphene membranes. While an engineering solution is presented, its novelty is not clearly articulated. The approach is highly tailored to the experimental setup, limiting its broader applicability, and may not necessarily represent the optimised reactor design for the author's system.

Some technical comments are below:

- It would be helpful to report the reaction kinetics (i.e., rate constants) in line 52.
- The boundary layer (line 54) needs to be quantified and its impact discussed. Distinct mass transfer regimes should be analysed—ideally supported by simulations and validated experimentally.
- It's important to differentiate between pore precursor nucleation, pore opening, and pore growth in the context of O₃ and light exposure etching. It's also important to state which of those steps the slit reactor used in this work is intended to modulate.
- Given this is fundamentally a mass transport study, claims about mass transfer enhancement (line 63) ought to be quantified.
- Several technical descriptions lack context and may be misleading. For example, line 75 states that oxidation at 80 °C achieves a gas-permeable pore density of 4 E11 cm⁻². It is unclear what oxidant is used and which gas is defined as "permeable." These details should be specified to avoid misinterpretation.
- Similarly, line 78 states, "oxidation rate is a function of temperature-dependent first-order kinetics." Is this a general claim or one specific to this experimental system?
- Critical experimental parameters are missing, including: chamber pressure, whether a pressure gradient exists, and whether a sweep gas is used during pore incorporation.
- The Reynolds number (Re) is central to the analysis. Viscosity, hydraulic diameter, and other parameters used to calculate Re should be reported. Re values of ~7–11 (line 104) seem low—are these for with or without the slit channels? The boundary layer thickness and Péclet number (Pe) should also be provided.
- Consider whether a simpler alternative—such as positioning the Cu foil further downstream from the leading edge of a stage in an open reaction tube and increasing the flow rate—could achieve comparable mass transfer enhancement without requiring a micro-slit chamber. Can such an approach offer greater practicality and scalability for implementation.
- The ~2× discrepancy between pore density measurements from Raman and AC-HRTEM needs to be explained. What are the uncertainties in the reported values? Given that pore density is central to the manuscript's main claim, this ought to be quantified.
- The pore density appears to plateau at $v = 45 \text{ cm s}^{-1}$ (lines 137–138). What's limiting the system at this point? Why does increasing velocity further not increase pore density?

- 80 keV electron beam irradiation may damage oxidized graphene. Are the authors confident that AC-HRTEM imaging does not induce further pore formation or growth after O₃ treatment? Since Figure 4 and later discussions rely on CO₂-permeable nanopores identified via TEM, this should be addressed.
- The number of AC-HRTEM images and total graphene area analyzed to produce the statistics in Figure 4c should be reported.
- The criteria used to classify pores (e.g., Pore-11 and above) as CO₂-permeable should be defined.
- Since O₃ chemisorbs on a finite surface area of the graphene, the key goal under consideration is pore density, not necessarily reaction rate. Could extended exposure time (beyond 1 hr) increase pore density?
- The resistance model used to calculate permeance in Figure 6 and Supplementary Note 2 is not clearly presented. The flow resistance of each component of the membrane composite and the these resistance values' reproducibility across membranes should be evaluated and reported.
- The reported CO₂/N₂ separation performance (e.g., <20 selectivity at ~1000 GPU CO₂ permeance) should be benchmarked against commercial and other membranes reported in the literature to position the results within the broader field.

Reviewer #3

(Remarks to the Author)

Built upon their previous work, the authors reported a further step that they have taken to produce high density of nanopores on single layer graphene using O₃ at room temperature. A micro channeled flow reactor was employed to reduce concentration polarization and increase the O₃ concentration on the surface of graphene, thus creating more epoxy functionalities and leading to higher density of pores. The approach is interesting. Some data and explanations in the manuscript are not complete or not fully convincing.

1. The authors claimed their method to be scalable. However, the method involves the use of micro-channeled flow reactor, which is not a scaled device.
2. Fig. 1, how did the authors fix the graphene sheet inside reactor? Does it move under O₃ purging?
3. CFD simulations were employed to explain the impact of micro-channels. If velocity/concentration is a main factor, why not directly using a normal device by increasing the O₃ flow rate? Or one can simply apply a O₃ stream with a higher O₃ concentration.
4. Why does the 2nd O₃ treatment mainly expand the vacancies?
5. The graphene after 2nd post treatment showed impressive CO₂ permeance and reasonable selectivity. How about mixed gas performance and the performance in the presence of humidity? These are important to evaluate the transport across graphene and the potential of graphene for CO₂/ N₂ separation.
6. The PTMSP supporting film has a serious aging problem. How does the graphene membrane perform over long term gas separation tests?

Version 1:

Reviewer comments:

Reviewer #1

(Remarks to the Author)

The authors have addressed all my concerns, and it can be accepted for publication.

Reviewer #2

(Remarks to the Author)

The authors have thoroughly addressed my previous questions, and I believe the manuscript is now suitable for publication. The transport analyses and the detailed description of pore characterization and pore size distribution are carefully conducted and will be valuable contributions for the advancement of the field. I am genuinely impressed by the quality of the work, and I strongly recommend its publication.

Reviewer #3

(Remarks to the Author)

The authors have addressed the technical comments. The drastic decrease of permeance in the presence of only 3% vapor implies that the membrane might not be quite feasible for CO₂/N₂ separation, though.

Response to the Reviewers' Comments

Reviewer 1:

I think the idea is good, and work is excellent. I would like to recommend the publication of this paper after the following few comments are addressed.

Author's response: We thank the reviewer for support and helpful comments, which have helped us to improve the quality of the manuscript.

1. I cannot find the novelty of this work. Similar type of work has been already published by same authors (<https://www.nature.com/articles/s44286-025-00203-z#Sec15>)

Author's response: We thank the reviewer for this important comment on novelty and for drawing attention to the need for a more precise articulation of this study's novelty.

The previous work (*Nature Chemical Engineering*, 1-11, 2025) relied on pore formation at elevated temperatures (~80-95 °C) for 1 h. No significant oxidation was observed in this work when oxidation was carried out at room temperature.

The current study introduces a more practical and scalable approach: room temperature oxidation and pore incorporation. Notably, this was done without increasing the reaction time (i.e., 1 h).

We developed a novel and simple slit-based oxidation reactor to significantly improve mass transfer. Specifically, we identify a range of velocities (of the order of 10-50 cm/s) that enhance reaction kinetics to the extent that oxidation could be achieved at room temperature. This is significant for the scale-up of this technology, given that it will be much more challenging to design scaled-up heated oxidation zones involving ozone, where ozone has a short half-life under elevated temperature conditions. Heated ozone is also challenging to handle and imposes restrictions on chamber materials, including sealings. This study will allow future integration of room-temperature oxidation of graphene in a continuous roll-to-roll fabrication protocol by a simple laminar flow-based system.

We now highlight these distinctions explicitly in the revised manuscript's Introduction section with the following additions in lines (71-76):

"Herein, unlike prior approaches that rely on elevated temperatures (~80–90 °C)³⁶ for ozone-based oxidation, we achieve Å-scale pore incorporation entirely at room temperature. Notably, this is accomplished without increasing reaction time, achieving oxidation in a short duration of 1 h. This represents the first demonstration of scalable and controllable ozone-based oxidation of graphene under fully ambient conditions. The resulting centimeter-scale membranes exhibit attractive CO₂/N₂ separation performances."

2. It is suggested that authors shall added key results in abstract to reflect the worth of manuscript.

Author's response: We thank the reviewer for this constructive suggestion.

We have now revised the abstract to include key quantitative results highlighting membrane performance:

We overcome this bottleneck using a micro-channeled flow reactor that enhances mass transfer, accelerating the oxidation rate, leading to a tenfold higher pore density at room temperature. Centimeter-scale porous graphene is achieved, resulting in CO₂/N₂ selectivity up to 21 and CO₂ permeance up to 4050 gas permeation units.”

3. A minor comment on terminology, see J. Membr. Sci. 2010, 348, 346-352. Nature Communications, 2018, 9, 486., Flux should be in a unit of Lm²h⁻¹, permeance should be in a unit of Lm²h⁻¹bar⁻¹, permeability should be the permeance normalized with thickness. Usually flux is used for organic separation, etc. Therefore, use appropriate terminology in MS.

Author’s response: We thank the reviewer for this helpful observation. We agree that the term “ozone flux”, as used in the manuscript, can be confusing. It referred specifically to the mass transfer of ozone gas from the bulk gas phase of the flow reactor to the surface of graphene. It did not refer to membrane performance.

Indeed, for membrane performance, the correct term is permeance, typically expressed in units such as gas permeation units (GPU; 1 GPU = 3.35 × 10⁻¹⁰ mol m⁻²s⁻¹Pa⁻¹) for gas separation, while flux is more commonly used in liquid or organic separations.

To avoid confusion, we have carefully revised the manuscript to ensure that “ozone flux” and “gas permeance” are clearly and consistently distinguished. The terminology has been checked throughout the manuscript in accordance with the literature. We have used “permeance” exclusively to describe gas transport through the graphene membranes, consistent with standard practice in gas separation literature.

4. Please check the “High-temperature oxidation (e.g., at 80 °C).....

Author’s response: We thank the reviewer for pointing this out. We intended to emphasize that ozone oxidation in prior studies required elevated temperatures, significantly higher than room temperature, which introduces additional complexity and safety considerations. Therefore, we referred to this condition as high-temperature oxidation. To avoid confusion and better reflect the relative nature of these conditions, we have revised the sentences as follows:

Lines (10-11): "However, incorporating a high pore density has until now required elevated-temperature ozone oxidation, while oxidation at room temperature was found to be sluggish, limiting scalability”

Line (39): “Ozone oxidation at elevated temperatures (e.g., at 80 °C) was needed to achieve an attractive combination of gas permeance and selectivity”

5. It is recommended to improve introduction section. Relevant queries / questions of research should be addressed in the introduction and try to avoid the AI use.

Author’s response: We thank the reviewer for the constructive suggestion, which helps us to improve the manuscript quality. We have revised the Introduction to better articulate the motivation behind our study and the specific scientific questions we address. In line with the reviewer’s comments, we highlight the novelty of our approach, particularly the role of room-temperature oxidation in enabling controlled pore formation in graphene, emphasize how

oxidation conditions critically influence final membrane performance through modulation of the density of functional groups, and we include kinetic parameters.

We have added the following text to the introduction starting with line 45:

“The reaction of O₃ with graphene leading to pore formation occurs through a sequence of steps: chemisorption of epoxy as pore precursor, clustering of epoxy, and finally, pore opening at the center of the strained epoxy. The first step, epoxy chemisorption, involves a first-order reaction of O₃ with graphene.⁴⁰ The mobile epoxies, diffusing with an activation barrier of ~0.73 eV^{41,42} on the graphene lattice, coalesce into organized epoxy trimer clusters to minimize the net energy of the oxidized graphene.^{37,43} As cluster size increases, the highly strained cluster core experiences C-C bond cleavage, generating ether linkages.^{39,44} Subsequent energy input, e.g., light irradiation, gasifies the cluster core, resulting in pore opening at its centre.^{38,39} Achieving a high CO₂ permeance requires a high density of pores, and therefore, a high density of epoxy clusters, and by extension, a high density of epoxy. Therefore, ensuring a high chemisorption rate of epoxy is crucial.

However, the reaction for pore precursor nucleation was found to be markedly sluggish at room temperature.^{31,33} The elementary first-order reaction of ozone with graphene proceeds with an estimated rate constant of ~1.4 s⁻¹ and an activation energy of 0.66 eV.^{38,40} During this reaction, the accumulation of O₂ byproducts near the surface creates a concentration polarization layer, limiting the local O₃ concentration and slowing down oxidation kinetics. This highlights the need for mass transfer-enhancing strategies to achieve effective oxidation at ambient conditions”.

We note that the manuscript was not written using AI.

6. Figure 3b-d showed some contamination. Can authors explain the reason for it and how we can avoid it? Is there effect of these contaminations on results?

Author’s response: We thank the reviewer for this important observation. The contamination observed in Figure 3b–d originates from a brief atmospheric exposure of graphene during the sample transfer process, specifically taking place during a period when the sample is removed from the ozone reactor and placed inside the XPS sample holder. We used a custom-designed XPS sample holder to reduce the time of exposure to the atmosphere. After ozone treatment, the sample was transferred into the holder inside a glove box, and the holder was designed to maintain vacuum conditions. It features a cap that automatically releases upon insertion into the XPS chamber, reducing exposure to ambient air.

However, during the brief transition between removing the sample from the reactor and placing it into the glove box, the sample was briefly exposed to air, which likely led to some unavoidable surface contamination. This brief handling step can result in adsorption of water and atmospheric contaminations on graphene and graphitic surfaces (*Nature Materials* 12, 925–931, 2013). We have clarified this in the manuscript (lines 176–179):

“To minimize atmospheric contamination on the samples, the samples were transferred to a custom XPS sample holder in less than one minute and were maintained under vacuum until introduction into the XPS chamber.”

We note that the contamination is minor and does not affect our conclusions. We compare relative changes in the concentration of functional groups (e.g., C–O ratios) across samples subjected to different oxidation conditions. For XPS, all samples underwent identical sample preparation, handling, and transfer protocols, and the degree of contamination remained consistent across all samples. Therefore, the minor contamination does not bias the comparisons.

7. Authors shall also focus on XRD of materials in revised work

Author's response: While XRD is a powerful technique for analyzing many crystalline materials, it is not well suited for characterizing a 2D film in this work, a single layer of graphene. We emphasize that the membrane reported in this work is based on a single layer of graphene and not stacked graphene sheets. XRD is highly informative to understand the periodic stacking of 2D sheets; however, it is not relevant in this study.

For understanding the evolution of functional groups and defects, we have characterized graphene with Raman spectroscopy and XPS. Further, to understand the porous structure of graphene, we have used aberration-corrected HRTEM.

8. Authors explained that “Even 5 min of post-ozone exposure nearly doubled the ID/IG ratio from 1.10 to 2.25 (Figure 5b). Longer treatments (15–30 min) further broadened the D, G, and 2D bands, diminished 2D peak intensity, and produced a pronounced (D + D') band, signifying greater lattice disorder. Did authors checked further longer treatment?”

Author's response: We thank the reviewer for this important comment. In the manuscript, we investigated post-ozone exposure times up to 30 minutes. As shown in Figure 5b, increasing the duration from 5 to 30 minutes caused progressive broadening of the D, G, and 2D bands, emergence of a pronounced (D + D') band, and a drop in 2D peak intensity, indicating increasing disorder in the graphene lattice. However, we also observed that beyond 15 minutes, membrane selectivity declined, which we attributed to excessive pore enlargement and reduced molecular sieving capability.

To address the reviewer's question, we further extended the post-treatment to 2 hours and acquired Raman spectra (figure below). The result showed even greater broadening of D and G bands and a further decrease in 2D band intensity, consistent with a higher degree of disorder. These findings confirm that longer post-treatments degrade lattice order. We have now included the Raman spectra in Supporting Information as Figure S12.

Figure S12. Raman spectra of graphene oxidized with ozone using FC900 for 1 h, followed by photonic gasification, and post-ozone exposure of 2 h.

We have revised the main text as follows:

“Even 5 minutes of post-ozone exposure nearly doubled the I_D/I_G ratio from 1.10 to 2.25. Longer treatments (15 min to 2 h) further broadened the D , G , and $2D$ bands, diminished $2D$ peak intensity, and produced a pronounced $(D + D')$ band, signifying greater lattice disorder (Figure 5b, Figure S12).”

9. Did authors compared their results with literature?

Author’s response: We thank the reviewer for this helpful suggestion. In the manuscript, we compared the literature on ozone oxidation for introducing pores in graphene (Figure 4a). Specifically, we evaluated defect densities resulting from:

- (i) oxidation at various temperatures (21–43 °C) without flow confinement, followed by thermal or photonic gasification
- (ii) room-temperature oxidation with confined flow (FC900), followed by photonic gasification

Figure 4a: Comparison of the oxidative etching approach used in this study with the literature.

We describe this comparison in the following text:

The manuscript mentions this as follows:

“Remarkably, vacancy density in these room-temperature oxidation samples exceeds that reported for oxidation at elevated temperatures (Figure 4a), independent of whether thermal or photonic gasification was used. For instance, photonic gasification after 43 °C oxidation yielded a lower density of $2.5 \times 10^{12} \text{ cm}^{-2}$.”

We acknowledge that a comparison with membranes for CO₂/N₂ separation is also valuable. To address this, we have now included a comparison plot in the Supporting Information, comparing graphene membranes with representative state-of-the-art membranes for carbon capture.

“The combination of improved high CO₂ permeance and attractive CO₂/N₂ selectivity achieved here underscores the strong potential of single-layer graphene membranes and highlights the importance of scalable fabrication methods to enable their industrial adoption (Figure S17, Table S4).”

Figure S17. Gas separation performances of the state-of-the-art and commercial membranes for CO₂/N₂ separation

Table S4. Comparison of carbon capture performance of porous graphene membrane.

Membrane Type	Note	CO ₂ permeance (GPU)	CO ₂ /N ₂ selectivity (separation factor)	Reference
Porous Single Layer Graphene	FC300	2691	16.6	This work
	FC300, resistance model	4035	21	
Commercial membranes	(Gen 1) Polaris®	1000	50	6
	(Gen 2) Polaris®	2000	49	7

	Prism	161	37	8
Polymeric membranes	Pebax2533/PEG-b-PPFPA	3330	22	9
	PEG/NH2-MIL-53	3000	34	10
Facilitated transport membranes	Ionic liquid on graphene	4000	20	11
	Amine-incorporated polymer	1450	185	12

Reviewer 2:

The study addresses a mass transfer–limited challenge, rather specific to the authors' system for O₃-induced nanopore formation in monolayer graphene membranes. While an engineering solution is presented, its novelty is not clearly articulated. The approach is highly tailored to the experimental setup, limiting its broader applicability, and may not necessarily represent the optimised reactor design for the author's system.

Author's response: We thank the reviewer for the constructive feedback, which has allowed us to improve the manuscript.

Some technical comments are below:

1. It would be helpful to report the reaction kinetics (i.e., rate constants) in line 52.

Author's response: We thank the reviewer for this insightful suggestion. To address this, we have included kinetic parameters from a recent study (*J. Phys. Chem. C* 127, 22015–22022, 2023), which combines ab initio calculations with temperature-dependent XPS experiments to analyze the chemisorption of ozone on graphene. The study reports chemisorption activation energy of 0.66 eV with an estimated rate constant of $\sim 1.43 \text{ s}^{-1}$ at room temperature, corresponding to a chemisorption timescale of $\sim 0.7 \text{ s}$. This confirms that ozone can react with graphene at ambient conditions.

We have added the following sentence in the revised manuscript (Line 56-62):

"However, the reaction for pore precursor nucleation was found to be markedly sluggish at room temperature.^{31,33} The elementary first-order reaction of ozone with graphene proceeds with an estimated rate constant of $\sim 1.4 \text{ s}^{-1}$ and an activation energy of 0.66 eV.^{38,40} During this reaction, the accumulation of O₂ byproducts near the surface creates a concentration polarization layer, limiting the local O₃ concentration and slowing down oxidation kinetics. This highlights the need for mass transfer–enhancing strategies to achieve effective oxidation at ambient conditions".

2. It's important to differentiate between pore precursor nucleation, pore opening, and pore growth in the context of O₃ and light exposure etching. It's also important to state which of those steps the slit reactor used in this work is intended to modulate.

Author's response: We appreciate the reviewer's insightful comment highlighting the need to clarify the distinctions between pore precursor nucleation, pore opening, and pore growth. Slit was explicitly used to improve the mass transfer of ozone.

The study focused on enhancing the pore precursor (epoxy) chemisorption step. The slit reactor geometry enhanced ozone mass transfer, thereby improving the chemisorption kinetics of epoxy. This allowed us to form a high density of epoxy clusters where pores form.

We then remove the slit and incorporate pores at the site of epoxy clusters by photonic gasification.

For pore enlargement, we applied a 2nd cycle of ozone in the same slit reactor.

For clarification, we have revised the manuscript to explicitly state that the slit reactor is primarily intended to modulate the pore precursor nucleation step.

“The reaction of O₃ with graphene leading to pore formation occurs through a sequence of steps: chemisorption of epoxy as pore precursor, clustering of epoxy, and finally pore opening at the center of the strained epoxy. The first step, epoxy chemisorption, involves a first-order reaction of O₃ with graphene.⁴⁰ The mobile epoxies, diffusing with an activation barrier of ~0.73 eV^{41,42} on the graphene lattice, coalesce into organized epoxy trimer clusters to minimize the net energy of the oxidized graphene.^{37,43} As cluster size increases, the highly strained cluster core experiences C-C bond cleavage, generating ether linkages.^{39,44} Subsequent energy input, e.g., light irradiation, gasifies the cluster core, resulting in pore opening at its centre.^{38,39} Achieving a high CO₂ permeance requires a high density of pores, and therefore, a high density of epoxy clusters, and by extension, a high density of epoxy. Therefore, ensuring a high chemisorption rate of epoxy is crucial.

However, the reaction for pore precursor nucleation was found to be markedly sluggish at room temperature.^{31,33} The elementary first-order reaction of ozone with graphene proceeds with an estimated rate constant of ~1.4 s⁻¹ and an activation energy of 0.66 eV.^{38,40} During this reaction, the accumulation of O₂ byproducts near the surface creates a concentration polarization layer, limiting the local O₃ concentration and slowing down oxidation kinetics. This highlights the need for mass transfer-enhancing strategies to achieve effective oxidation at ambient conditions.”

For pore expansion step (line 68-70):

“A brief, room-temperature second cycle ozone exposure within the slit results in pore expansion, which further enhances membrane performance.”

3. Given this is fundamentally a mass transport study, claims about mass transfer enhancement (line 63) ought to be quantified.

Author’s response: We thank the reviewer for highlighting the need to support our claim regarding mass transfer enhancement quantitatively.

While direct experimental quantification of the mass transfer coefficient under our specific conditions is challenging, we provide quantitative evidence of mass transfer enhancement based on X-ray photoelectron spectroscopy (XPS) measurements.

In the manuscript, XPS data (Figure 3e) show an increase in C–O functional group content on graphene samples exposed to ozone under different flow conditions. Importantly, these measurements were obtained under identical ozone concentration, exposure time (1 h), and room temperature conditions. The only variable was the flow configuration, which changes the local gas velocity and hence the mass transfer rate. We have now included the following discussion on mass transfer enhancement in Supplementary Information as Note S3, Table S2, and Figure S6.

To relate XPS findings to classical mass transfer theory, we use the Sherwood number.

“Note S4. Estimation of mass transfer enhancement with increased O₃ flow velocity

Sherwood number shows the relation between convective mass transfer to diffusive mass transfer:

$$Sh = \frac{k_m L}{D}$$

where, k_m is the mass transfer coefficient, L is the characteristic length, D is the diffusivity.

For laminar flow between flat plates, the Sherwood number scales as ⁴:

$$Sh = 1.85 Re^{1/3} Sc^{1/3}$$

and for tube:

$$Sh = 1.62 Re^{1/3} Sc^{1/3}$$

$$Re = \frac{\rho V L}{\mu}$$

$$Sc = \frac{\mu}{\rho D}$$

ρ is fluid density, μ is the dynamic viscosity, and V is the velocity. Therefore, the mass transfer coefficient scales with gas velocity as:

$$k_m \propto V^{1/3}$$

Table S2. The relative increase in mass transfer coefficients based on the average velocity of different flow configurations and corresponding C-O% content

Condition	Velocity (cm/s)	C-O (%) from XPS	Relative k_m
w/o Flow Channel	0.5	~ 4.1	1
FC900	15	~ 8.2	3.1 fold
FC300	45	~ 13	4.5 fold

For a first-order surface reaction, surface flux can be defined as:

$$J_{O_3} = k_m * C_{O_3}$$

If each reaction event yields a C–O bond, assuming that the bulk ozone concentration is constant, then the total C–O content after a fixed time is:

$$[C - O] \propto \int J_{O_3} dt \propto V^{1/3}$$

By comparing the XPS-measured C–O content with $V^{1/3}$, the relative increase in mass transfer coefficient can be confirmed.

As shown in the plot below, the C–O content increases linearly with $V^{1/3}$. The strong linear correlation ($R^2 \sim 0.97$) confirms that enhanced gas velocity leads to increased ozone delivery

to the surface, thereby accelerating the surface oxidation process. The estimated ~3.1 and ~4.5 fold increase in mass transfer coefficients for FC900 and FC300, respectively, provides a quantitative basis for the observed enhancement in surface functionalization.

Figure S7. The change in C-O% content on graphene under different O₃ flow velocities

We also included this extended analysis in the manuscript (lines 184-190):

“Based on the $V^{1/3}$ scaling of the mass transfer coefficient, we estimate that the use of confined microchannel flow increases the ozone mass transfer coefficient by a factor of ~3.1 (FC900) and ~4.5 (FC300) compared to the unconfined condition. Correspondingly, the intensity of C-O peak increased with increasing mass transfer (Figure 3b-d), with the highest population of C-O (~13%) with the highest v (45 cm s⁻¹), consistent with the Raman data, confirming a significant enhancement in mass transport under these flow configurations (Supplementary Note S4, Table S2).”

4. Several technical descriptions lack context and may be misleading. For example, line 75 states that oxidation at 80 °C achieves a gas-permeable pore density of 4 E11 cm⁻². It is unclear what oxidant is used and which gas is defined as “permeable.” These details should be specified to avoid misinterpretation. Similarly, line 78 states, “oxidation rate is a function of temperature-dependent first-order kinetics.” Is this a general claim or one specific to this experimental system?

Author’s response: We thank the reviewer for this important point of avoiding ambiguity, which certainly helps to improve the quality of the manuscript. We have now clarified the sentence on line 75 (now line 84 in revised text) to specify both the oxidant and the target gas. Specifically, we now state that the reported pore density of 4×10^{11} cm⁻² was achieved via ozone (O₃) oxidation at 80 °C, and the resulting pores were characterized to be CO₂-permeable.

“For example, a CO₂-permeable pore density of 4×10^{11} cm⁻² could be obtained using O₂/O₃ gas mixture as the oxidizing agent at a temperature of 80 °C.³⁹ As we detail below, oxidation using ozone at room temperature yields negligible pore density under conventional conditions.”

The statement in line 78 (now line 88) refers specifically to the oxidation of graphene by O₃, consistent with prior experimental and theoretical studies. We revised the sentence as follows:

“Ozone oxidation rate is a function of temperature-dependent first-order kinetics, with a rate constant of $\sim 1.4 \text{ s}^{-1}$ at room temperature.⁴⁰”

5. Critical experimental parameters are missing, including: chamber pressure, whether a pressure gradient exists, and whether a sweep gas is used during pore incorporation.

Author’s response: We thank the reviewer for this helpful comment. We clarify the following key experimental parameters related to the pore incorporation step:

- **Chamber pressure:** All oxidation experiments were conducted at atmospheric pressure (1 bar).
- **Pressure gradient:** No external pressure gradient was applied across the graphene samples during oxidation. The flow was confined but not pressurized.
- **Sweep gas:** No sweep gas was used. The ozone feed consisted of a dilute mixture of ozone in oxygen (9.35 mol% O₃ in O₂) generated from an ozone generator.

We revised our Methods section, including these parameters:

“These flow channels were inserted into a tubular quartz reactor maintained at ambient pressure (~ 1 bar) and connected to an ozone generator (Absolute Ozone, Atlas 60). For oxidation, samples were exposed to of 100 sccm of a 9.35 mol% O₃ / O₂ gas mixture for 1 hour at room temperature (no additional sweep gas was used). No pressure gradient was applied across the graphene during this process.”

6. The Reynolds number (Re) is central to the analysis. Viscosity, hydraulic diameter, and other parameters used to calculate Re should be reported. Re values of ~ 7 –11 (line 104) seem low—are these for with or without the slit channels? The boundary layer thickness and Péclet number (Pe) should also be provided. The boundary layer (line 54) needs to be quantified and its impact discussed. Distinct mass transfer regimes should be analysed—ideally supported by simulations and validated experimentally.

Author’s response: We thank the reviewer for this helpful comment, which helps to improve the manuscript. Below, we provide the parameters for Reynolds number estimation and also report Péclet number and boundary layer thickness for different flow configurations based on the following equations and COMSOL simulations. We included this analysis in Supporting Information as Note S2, Figure S2, and Table S1.

“**Note S2. Reynolds number, Peclet number, and boundary layer thickness estimation of different flow configurations**

Assuming diluted inlet stream as 100 sccm pure O₂ at 25 C, 1bar; ρ is fluid density (1.3 kg/m³), μ is the dynamic viscosity (2×10^{-5} Pa.s), and V is the average velocity (0.005, 0.09, 0.3 m/s for w/o flow channel, FC900 and FC300, respectively), H is hydraulic diameter (2, 0.172, and

0.059 cm, for w/o flow channel, FC900 and FC300, respectively), and D is diffusivity ($2 \times 10^{-5} \text{ m}^2 \text{ s}^{-1}$), Reynolds and Peclet numbers can be calculated by following equations:

$$Re = \frac{\rho V H}{\mu}$$

$$Pe = \frac{V H}{D}$$

CFD simulations with velocity profiles, for cross-section of the geometry, of various flow configurations are shown below. Profile lines for each flow configuration are indicated with red dashed lines. Boundary layer thickness is estimated as the distance where velocity is reached to 99% of the maximum velocity in the reactor.

Figure S2. CFD simulations of different flow configurations showing velocity profiles.

Table S1. Reynolds number, Peclet number, and boundary layer thickness in various flow configurations.

Condition	Re	Pe	δ (mm)
w/o Flow Channel	7	5	~ 4.3
FC900	10.7	90	~ 0.4
FC300	11.5	300	~ 0.12

We have now included these values in the main text (lines 130-135):

“The profiles show that FCs drastically increases the average v near the surface, rising from 0.5 cm s^{-1} (without a flow channel) to 15 cm s^{-1} (with FC900) and further increasing to 45 cm s^{-1} (with FC300). Correspondingly, the boundary layer thickness decreases from $\sim 4.3 \text{ mm}$ (without a flow channel) to $\sim 0.12 \text{ mm}$ (with FC300). This transition is captured by the Péclet number, which increases from ~ 5 to ~ 300 , indicating a shift from diffusion-limited to convection-enhanced transport. Such enhancement is expected to improve the mass transfer of ozone to the graphene surface (Supplementary Note S2, Table S1).”

7. Consider whether a simpler alternative—such as positioning the Cu foil further downstream from the leading edge of a stage in an open reaction tube and increasing the flow rate—could achieve comparable mass transfer enhancement without requiring a micro-slit chamber. Can such an approach offer greater practicality and scalability for implementation.

Author’s response: We are highly appreciative of the reviewer’s insightful suggestion regarding alternative strategies for enhancing mass transfer.

Indeed, placing the graphene sample further downstream in an open reaction tube and raising the overall flow rate can, in principle, reduce the diffusion boundary layer thickness and improve local ozone delivery. However, there are several practical limitations to this approach. We show that achieving a velocity $15\text{--}45 \text{ cm/s}$ (in FC900 and FC300, respectively) is essential to improve mass transfer, compared to $\sim 0.5 \text{ cm/s}$ in a tube configuration. Using slits, we could use a manageable ozone flow rate of 100 sccm . This increase in local velocity arises from geometric confinement and leads directly to enhanced mass transfer to the graphene surface.

To achieve comparable velocities in a tube without slit (2 cm diameter, as used in our experiments), one would require flow rates in the range of ~ 3000 to 9000 sccm . This would require 30 to 90 times higher ozone consumption, greater handling complexity, increased safety risks, and elevated operational costs. These challenges become even more pronounced at larger scales, where the risks and costs associated with handling high volumes of reactive gases are amplified.

An additional issue is that ozone is typically obtained by ozone generators, where a tradeoff between flow rate and concentration of ozone exists. This implies that generally, the concentration of ozone falls at higher flow rates. This would again compromise the reaction kinetics.

Overall, while the reviewer's proposed approach could conceptually improve mass transfer, the flow channel offers a practical solution for achieving high local velocities and enhanced oxidation kinetics with a manageable flow rate of ozone.

8. The $\sim 2\times$ discrepancy between pore density measurements from Raman and AC-HRTEM needs to be explained. What are the uncertainties in the reported values? Given that pore density is central to the manuscript's main claim, this ought to be quantified.

Author's response: We thank the reviewer for this important observation. The discrepancy between the pore densities estimated from Raman spectroscopy and those measured via AC-HRTEM arises from the fundamental differences between the two techniques.

Defect densities derived from Raman spectroscopy rely on the carbon amorphization trajectory model, which relates the I_D/I_G ratio to the average inter-defect distance L_D and subsequently to the defect density. While this method is well-established and useful for tracking trends across a broad range of samples, it has known limitations in the high-defect-density regime. In particular, as the graphene structure transitions toward nanocrystalline, the D -band intensity begins to saturate or decrease, leading to an underestimation of the true defect or pore density.

To address this, we do not rely solely on Raman spectroscopy to quantify pore density. Instead, we use Raman to monitor the relative evolution of disorder across different oxidation conditions, while using AC-HRTEM as a complementary technique for direct, high-resolution imaging of atomic-scale vacancies and pores. AC-HRTEM enables us to count individual defects and therefore provides a more accurate estimation of defect density.

9. The pore density appears to plateau at $v = 45 \text{ cm s}^{-1}$ (lines 137–138). What's limiting the system at this point? Why does increasing velocity further not increase pore density?

Author's response: Based on AC-HRTEM images, the pore density increased from 0 (without flow channel) to $1.4 \times 10^{11} \text{ cm}^{-2}$ (FC900) and then $2.2 \times 10^{11} \text{ cm}^{-2}$ (FC300), with increasing velocity near the graphene surface.

The oxidation process determines the pore density on the sample, as defects originate from the gasification of oxygen-containing functional groups formed during O_3 exposure. Therefore, the pore density is proportional to the density of surface C–O functional groups, which itself scales with the surface ozone flux and the mass transfer coefficient k_m . Therefore, we should track changes in the pore density based on $V^{1/3}$ in a given flow configuration.

We quantified the relative enhancement in the mass transfer coefficient based on velocity scaling ($k_m \propto V^{1/3}$) and found that k_m increases by a factor of 3.1 fold for FC900 and 4.5 fold for FC300, relative to the case where we do not use the flow channel. When we plot pore density against this relative k_m , we observe a strong linear correlation (R^2 close to 1), which shows that with increased velocity, pore density increases:

Figure S11. Change in the pore density with relative mass transfer coefficient (estimated based on $k_m \propto V^{1/3}$).

We have now included this analysis in Supporting Information as Figure S11, and also referred to it in the main text

"However, pore density increased significantly with increasing average velocity, as indicated by the increase in the relative mass transfer coefficient, which scales as $k_m \propto V^{1/3}$ (Supplementary Note S4, Figure S11)."

10. 80 keV electron beam irradiation may damage oxidized graphene. Are the authors confident that AC-HRTEM imaging does not induce further pore formation or growth after O_3 treatment? Since Figure 4 and later discussions rely on CO_2 -permeable nanopores identified via TEM, this should be addressed.

Author's response: We agree that the 80 keV electron beam has sufficient energy to drive the gasification of oxygen-containing functional groups on graphene. Previous studies (e.g., *J. Am. Chem. Soc.*, 133, 17315–17321, 2011, *Advanced Materials* 34, no. 51, 2206627, 2022) have shown that the energy barrier for the gasification of oxygen clusters into CO or CO_2 is approximately 1.1-1.3 eV, which can be exceeded under high-energy electron beam irradiation.

To mitigate this and to ensure that the observed nanopore structures reflect the state of the skeleton of the pore structure after ozone treatment and not beam-induced artifacts, we subjected the samples to hydrogen annealing at 600 °C for 1 h prior to imaging. This approach was guided by earlier work on HRTEM imaging of graphene pores, which demonstrates that H_2 annealing effectively removes oxygen-containing functional groups from the graphene lattice without introducing new pores or significantly expanding existing ones. (*Science Advances* 7, eabf0116, 2021; *ACS Nano* 15, 13230-13239, 2021).

We included this detail in the Methods section:

"Sample preparation for imaging was explained in our previous work.^{33,57} Before imaging, TEM grids were cleaned in H₂ environment at 600 °C for 1 h."

11. The number of AC-HRTEM images and total graphene area analyzed to produce the statistics in Figure 4c should be reported.

Author's response: For the pore density statistics presented in Figure 4c, we acquired a total of 69 AC-HRTEM images across multiple regions of each graphene sample. The total imaged area per sample was approximately 25200–31500 nm². For the pore size distribution (PSD) analysis, only clearly resolved vacancies were included in the pore count, and we further restricted the dataset to pores not overlapping with contaminations to avoid artifacts in size estimation. These details are added to the Methods section:

"For defect density and pore size distribution analysis, a total area of approximately 25200–31500 nm² per sample was analyzed from high-resolution AC-HRTEM images. For PSD calculations, only pores not intersecting contamination were included to ensure accurate size estimation."

12. The criteria used to classify pores (e.g., Pore-11 and above) as CO₂-permeable should be defined.

Author's response: We thank the reviewer for raising this important point, which helps the clarity of the manuscript. We classified Pore-10 and above as CO₂-permeable, based on pore limiting diameter estimations and molecular dynamics simulations in the literature (*Nature Communications* 16, 6252, 2025). Pore-10 has a PLD of ~2.7 Å, which is close to the kinetic diameter of CO₂ (3.3 Å). Pores of this size, especially when decorated with O-functional groups, exhibit dynamic opening behavior, enabling selective CO₂ permeation through the flip-flop motion of terminal semiquinone groups. In the manuscript, we referred to this work (line 223-225):

For better clarity, we revised the manuscript text as follows:

"To evaluate the density of CO₂-permeable pores, defects were classified by the number of missing carbon atoms, e.g., a defect missing ten carbon atoms was labeled pore-10. Based on literature on CO₂-transport from functionalized graphene pores,⁵² only pores with size equal to or larger than pore-10 are CO₂-permeable; smaller pores were classified as impermeable due to insufficient electron density gap for CO₂ permeation."

13. Since O₃ chemisorbs on a finite surface area of the graphene, the key goal under consideration is pore density, not necessarily reaction rate. Could extended exposure time (beyond 1 hr) increase pore density?

Author's response: We have noticed that increasing the reaction time indeed increases the density of the functional group and pore density. To investigate the effect of extended ozone exposure time on pore density, we exposed graphene samples to a 100 sccm O₂/O₃ mixture at room temperature using the FC900 microchannel reactor for 1 and 2 h, followed by photonic

gasification. We fabricated two centimeter-scale membranes for each condition (labeled as M1 and M2) and evaluated their gas separation performance.

As shown in the figure below, increasing the O₃ exposure time from 1 to 2 h led to an increase in CO₂ permeance, suggesting a higher pore density, consistent with the reviewer's hypothesis. However, this increase in permeance was accompanied by a decrease in CO₂/N₂ selectivity, indicating the possible coalescence of nearby functional groups or the formation of larger epoxy clusters. This behavior implies that prolonged oxidation can compromise pore size control, leading to broader distributions and reduced selectivity.

The new experimental results have been added to the Supporting Information and are referenced in the revised manuscript text accordingly.

In Supporting Information:

Figure S14. Gas separation performance of porous graphene fabricated using FC900, prepared using oxidation times of 1 and 2 h.

In the main text, lines (318-321):

"When the oxidation duration was increased from 1 to 2 h using FC900, CO₂ permeance further increased above 2000 GPU (Figure S14). However, this was accompanied by a decrease in CO₂/N₂ selectivity, indicating the formation of larger pores likely due to coalescence of nearby pores."

14. The resistance model used to calculate permeance in Figure 6 and Supplementary Note 2 is not clearly presented. The flow resistance of each component of the membrane composite and the these resistance values' reproducibility across membranes should be evaluated and reported.

Author's response: We thank the reviewer for pointing out the need to clarify the resistance model used in our permeance calculations.

In Figure 6, gas permeance was analyzed using a resistance model, where the total membrane resistance is considered as the sum of the individual resistances of each layer. This is explained in Note S5 as follows:

Note S5. Calculation of graphene layer permeance

Gas permeance of the porous graphene layer was determined by the resistance model, using the measured stack layer permeance values. The resistance (R) of the membrane through gas transport can be defined as:

$$R = \frac{1}{J \cdot A}$$

A represents the actual membrane area, and J is the membrane flux.

The total resistance (R_T) of the graphene membrane is the sum of the resistances of layers ; graphene, PES, and PTMSP supports.

$$R_{total} = R_{PES} + R_{graphene} + R_{PTMSP}$$

PES layer resistance can be neglected since it is highly permeable, accordingly, graphene layer flux can be calculated by the following equation :

$$J_{graphene} = \frac{J_{total} \cdot J_{PTMSP}}{J_{total} - J_{PTMSP}}$$

To evaluate the contribution and reproducibility of each component, we fabricated and tested multiple centimeter-scale membranes with identical configurations. Table S3 (now included in the Supporting Information) reports the CO_2 and N_2 permeances (in GPU) and CO_2/N_2 selectivities across different membranes.

Table S3. CO_2 and N_2 gas permeances (in GPU) and CO_2/N_2 selectivities of centimeter scale membranes

(PTMSP CO_2 Permeance: 11125 ± 625 GPU, CO_2/N_2 Selectivity : 6.9 ± 0.6^5)

	PTMSP/Gr/PES			Graphene Layer		
	CO_2 (GPU)	N_2 (GPU)	CO_2/N_2	CO_2 (GPU)	N_2 (GPU)	CO_2/N_2
w/o FC	75	-	-	75.5	-	-
w/o FC	53	-	-	53.3	-	-
w/o FC	62	-	-	62.3	-	-
FC900	1407	79	17.6	1610	84	19.1
FC900	1395	88	15.8	1595	93	17
FC900	1610	100	16.1	1882	106	17.7
FC900	1490	88	17	1720	92.3	18.6
FC900	1210	80	15.3	1357	83	16.3
FC900_2h	1950	154	12.6	2364	171	13.8

FC900_2h	1620	114	14.1	1896	123	15.3
FC300	2345	128	18.3	2971	139	21.4
FC300	2691	178	16.6	4035	200	20.1
FC300	2557	213	12	3320	245	13.5
FC900_5 min	2419	155	15.6	3091	171	18
FC900_5 min	2700	180	15	3565	202	17.6
FC900_5 min	2972	188	16	4055	209	19.3
FC900_5 min	2750	200.8	13.7	3652	229	15.9
FC900_5 min	2870	215.8	13.3	3868	249	15.5
FC900_15 min	2883	171	16.8	3891	192	20.3
FC900_15 min	2872	159	18	3871	177	21.9
FC900_15 min	2740	161	17	3635	179	20.3
FC900_30 min	4150	518	8	6620	761	8.7
FC900_30 min	5410	688	7.9	10531	1193	8.8

15. The reported CO₂/N₂ separation performance (e.g., <20 selectivity at ~1000 GPU CO₂ permeance) should be benchmarked against commercial and other membranes reported in the literature to position the results within the broader field.

Author's response: We thank the reviewer for this important suggestion. As shown in Table S3 (see the table in response to comment #14), the performance of the porous graphene layer is relatively high, marked by permeance reaching 4000 GPU and selectivity above 20 for our best condition (FC900_15 min). The performance of the membrane is limited by the resistance of the polymeric support film, which is about 1 μm thick in this study. This led to an average membrane permeance of 2832 GPU, and a corresponding CO₂/N₂ selectivity of 17.3.

As also recommended by Reviewer 1, we prepared a benchmarking plot (now included in the Supporting Information) to compare porous graphene membrane with promising carbon capture membranes reported in the literature.

We note that for carbon capture, a higher permeance membrane is advantageous. Future work will improve membrane performance by reducing the support film resistance (reducing the thickness of the support film from 1 μm to down to 100-200 nm), and by pyridinic nitrogen functionalization of graphene pores (Nature Energy, 9, 964-974, 2024).

Figure S17. Gas separation performances of the state-of-the-art and commercial membranes for CO₂/N₂ separation

The list of membranes is also shown in the Supplementary Information as a table :

Table S4. Comparison of carbon capture performance of porous graphene membrane.

Membrane Type	Note	CO ₂ permeance (GPU)	CO ₂ /N ₂ selectivity (separation factor)	Reference
Porous Single Layer Graphene	FC300	2691	16.6	This work
	FC300, resistance model	4035	21	
Commercial membranes	(Gen 1) Polaris®	1000	50	6
	(Gen 2) Polaris®	2000	49	7
	Prism	161	37	8
Polymeric membranes	Pebax2533/PEG-b-PPFPA	3330	22	9
	PEG/NH ₂ -MIL-53	3000	34	10
Facilitated transport membranes	Ionic liquid on graphene	4000	20	11
	Amine-incorporated polymer	1450	185	12

Reviewer 3:

Built upon their previous work, the authors reported a further step that they have taken to produce high density of nanopores on single layer graphene using O₃ at room temperature. A micro channelled flow reactor was employed to reduce concentration polarization and increase the O₃ concentration on the surface of graphene, thus creating more epoxy functionalities and leading to higher density of pores. The approach is interesting. Some data and explanations in the manuscript are not complete or not fully convincing.

Author's response: We thank the reviewer for the constructive and helpful comments.

1. The authors claimed their method to be scalable. However, the method involves the use of micro-channelled flow reactor, which is not a scaled device.

Author's response: We thank the reviewer for raising a very important point of scalability. We think the term "micro-channel" is confusing the scalability aspect. Our flow channel are 300 μm and 900 μm wide, i.e., 0.3 mm and 0.9 mm wide. These macroscopic length scales can be designed in a scale-up reactor using conventional mechanical construction tools.

An important and central aspect of our study is highlighting the velocity needed to improve the mass transfer, so that the reaction can be achieved at room temperature. These velocities are identified in the order of 10 cm/s (15 and 45 cm/s). In future scale-up studies, one can also use alternate geometries (e.g., showerheads) to access these velocities.

2. Fig. 1, how did the authors fix the graphene sheet inside reactor? Does it move under O₃ purging?

Author's response: The top side of the slit reactor is designed with side legs that secure samples with a width of 2 cm when assembled on top. These legs not only hold the samples in place but also provide the defined flow gap. The side profile of the flow channel is shown below.

Figure S1. Side view of flow channel

We further noticed that even smaller samples (for example, dimension of 1.5 x 6 cm², weight of 0.46 g) also lie flat inside the reactor slit and remain stable during ozone flow.

A horizontal gas velocity of 50-90 cm/s generates a drag force ($\sim 10^{-7}$ N) that is over four orders of magnitude smaller than the gravitational force ($\sim 10^{-2}$ N) and frictional resistance ($\sim 10^{-3}$ N). Thus, the foil experiences neither lifting nor sliding during treatment. Experimental observation

confirmed no displacement. We included this in Supporting Information as Figure S1 and Note 1, and mentioned in the main text.

Below, we provide a detailed calculation for the above:

Note S1. Mechanical stability analysis of graphene samples inside slit flow reactor

The forces on the graphene coupon can be estimated using following equations:

$$F_{drag} = \frac{1}{2} \rho v^2 A C_d$$

where ρ is gas density ($\sim 1.3 \text{ kg m}^{-3}$ for O_2 at $25 \text{ }^\circ\text{C}$), A is the frontal area of Cu supported graphene sample exposed to flow ($1.5 \text{ cm} \times 100 \text{ }\mu\text{m}$), C_d is drag coefficient (~ 1.2 for a flat plate in laminar flow), v is velocity of ozone ($15\text{-}45 \text{ cm s}^{-1}$) near the graphene surface in the flow channel.

For $1.5 \times 6 \text{ cm}^2$ graphene sample placed in FC300 reactor ;

$$F_{drag} = \frac{1}{2} (1.3 \text{ kg m}^{-3}) \times (0.45 \text{ m s}^{-1})^2 \times (1.5 \times 10^{-6} \text{ m}^2) \times 1.2 = \sim 2.1 \times 10^{-7} \text{ N}$$

Given that the mass (m) of a $1.5 \times 6 \text{ cm}^2$ graphene coupon is 0.46 g ,

$$F_{gravity} = mg = 0.46 \text{ g} \times 9.8 \text{ m s}^{-2} = \sim 4.5 \times 10^{-3} \text{ N}$$

If we take into account the frictional force between Cu support and stainless steel with friction constant $\mu = 0.4$;

$$F_{friction} = \mu \times F_{gravity} = \sim 1.8 \times 10^{-3} \text{ N}$$

F_{drag} is almost negligible compared to frictional resistance, ensuring the graphene sample remains fixed under treatment conditions.

In the main text:

"The top side of the slit designed with two sides legs that secure samples with a width of 2 cm when assembled on top. These legs not only hold the samples in place but also provide the defined flow gap. (Figure S1) Considering the thickness of Cu foil ($\sim 100 \text{ }\mu\text{m}$), this resulted in a flow channel (FC) gap of 900 and $300 \text{ }\mu\text{m}$, respectively. Accordingly, these slits are termed as FC900 and FC300, respectively. Samples smaller than 2 cm size also remained fixed inside the slit at varying flow rates, due to high frictional resistance (Supplementary Note S1)."

3. CFD simulations were employed to explain the impact of micro-channels. If velocity/concentration is a main factor, why not directly using a normal device by increasing the O_3 flow rate? Or one can simply apply a O_3 stream with a higher O_3 concentration.

Author's response: We appreciate the reviewer's thoughtful comment regarding the rationale for using micro-channel structures instead of simply increasing the total O_3 flow rate or applying a higher O_3 concentration. While velocity and local concentration are indeed key parameters in surface reactions, there are several critical practical limitations:

First, the maximum achievable ozone concentration is limited by the performance of the ozone generator. Increasing the flow rate of oxygen through the generator reduces the residence time

of oxygen in the generator, thereby lowering the conversion efficiency and resulting in a decreased O₃ partial pressure. Therefore, one sacrifices ozone concentration at higher flow rates. Details of performance tests can be found in ozone generator supplier website : <https://absoluteozone.com/industrial-ozone-generators/magnum-120/>

Second, to achieve a comparable velocity in a tube without slit (2 cm diameter, as used in our experiments), one would require flow rates in the range of ~3000 to 9000 sccm. This would require 30 to 90 times higher ozone consumption, greater handling complexity, increased safety risks, and elevated equipment and operational costs. These challenges become even more pronounced at larger scales, where the risks and costs associated with handling high volumes of reactive gases are amplified. Considering future scale-up, significantly increasing flow rate becomes also a very expensive approach.

Third, increasing the total volumetric flow rate to increase velocity can lead to elevated back pressure, can destabilize flow, compromise seal integrity (safety), and can affect the stable positioning of graphene foils. Our micro-channel approach overcomes these limitations by geometrically confining the flow to enhance local velocity without increasing the total flow rate. With a fixed input of 100 sccm, we achieve near-surface velocities of ~15 cm/s (FC900) and ~45 cm/s (FC300), compared to ~0.5 cm/s in a tube configuration.

4. Why does the 2nd O₃ treatment mainly expand the vacancies?

Author's response: We thank the reviewer for raising an important point. The second cycle of O₃ exposure primarily expands pre-existing pores rather than inducing new ones in the graphene lattice. We addressed this point in the manuscript (lines 245–247), supporting our claim with HRTEM analysis by comparing the defect density before and after the second O₃ treatment cycle. The results show that the defect density remained comparable (3.2×10^{12} vs. $3.5 \times 10^{12} \text{ cm}^{-2}$) after 15 min of 2nd O₃ cycle, indicating no significant formation of new pores upon 2nd O₃ treatment.

At room temperature, the energy barrier for creating new pores inside a newly formed epoxy cluster is too high (1.1-1.3 eV) to overcome. In contrast, existing pores are significantly more reactive toward O₃. In our previous work (*Advanced Functional Materials*, 2503121, 2025), we demonstrated that when graphene rests on a Cu substrate, electron puddles formed by charge transfer locally dope the graphene. This doping lowers the energy barrier for gasification near the pore edges upon O₃ exposure. This creates a favorable condition for the selective expansion of existing pores.

To highlight this, we have added the following text to the manuscript:

“We therefore applied a second O₃ cycle at room temperature, exploiting the reactivity differences for C at and away from pore edges.⁵³ At room temperature, the energy barrier (1.1-1.3 eV) for creating new pores inside epoxy clusters is too high to overcome. In contrast, existing pores, resting on Cu foil, are significantly more reactive toward O₃. This is because when graphene rests on a Cu substrate, electron puddles formed by charge transfer locally dope the graphene. This doping lowers the energy barrier for gasification near the pore edges upon O₃ exposure. This creates a favorable condition for the selective expansion of existing pores.”

5. The graphene after 2nd post treatment showed impressive CO₂ permeance and reasonable selectivity. How about mixed gas performance and the performance in the presence of humidity? These are important to evaluate the transport across graphene and the potential of graphene for CO₂/ N₂ separation.

Author's response: We thank the reviewer for this important question regarding the realistic performance of the graphene membranes under mixed-gas and humid conditions.

To evaluate practical performance, we prepared a membrane using the FC900 configuration, with an oxidation time of 1 h. After photonic gasification to open pores, the membranes were subjected to a second ozone exposure (5 min).

Gas separation performance was then measured under: (i) dry mixed-gas conditions (equimolar CO₂/N₂), and (ii) humidified mixed-gas conditions (same gas ratio, with 3 mol% water vapor in the feed).

Prior to testing, the membrane was thermally annealed at 130 °C under CO₂ flow to remove surface contaminants, following which it was cooled to room temperature.

We observe parity performance for dry mixture feed. In the presence of water vapor, the CO₂ permeance dropped by ~35% relative to dry conditions. However, the CO₂/N₂ selectivity increased by ~50%. This is likely because water is expected to adsorb on O-functionalized graphene pores, where it would compete for transport, especially against N₂.

We included these tests in Supporting Information, and refer them in the main text:

Figure S16. Gas mixture separation performance of porous graphene membrane prepared by 1 h oxidation in FC900 followed by 5 min 2nd cycle of ozone, under dry gas mixture (equimolar CO₂ and N₂) and humidified mixture (equimolar CO₂ and N₂ with 3% water vapor) feed. Initial performance with dry stream taken as reference.

“Graphene membrane prepared by FC900 followed by 2nd cycle of O₃ oxidation was also tested with mixed gas feed (equimolar CO₂ and N₂) and humidified feed. When 3% water vapor was present in the feed, CO₂ permeance dropped by ~35% relative to dry conditions, but interestingly CO₂/N₂ selectivity increased by ~50% (Figure S16).”

6. The PTMSP supporting film has a serious aging problem. How does the graphene membrane perform over long term gas separation tests?

Author’s response: We thank the reviewer for raising an important point. We agree that PTMSP is prone to physical aging, which typically results in a decline in gas permeance over time due to densification of the polymer matrix and loss of free volume. This is also one of the reasons we used the resistance model to more accurately isolate and evaluate the intrinsic performance of the graphene layer.

To assess long-term stability, we tested a centimeter-scale graphene membrane under continuous CO₂ flow. The membrane was fabricated using the FC900 flow channel, with an ozone exposure time of 1 h. The initial CO₂ permeance and CO₂/N₂ selectivity were 1500 GPU and 16, respectively. Over 60 hours of operation, we observed a ~15% reduction in CO₂ permeance (Figure R2).

To investigate performance recovery, the membrane was simply heated at 130 °C in a pure CO₂ atmosphere for 2 hours. This treatment was intended to relax the polymer matrix and expand the polymer chains that had compacted over time, thereby increasing the free volume. Following this regeneration step, the CO₂ permeance was partially recovered.

To mitigate the aging issue, alternative support films such as those made of nanoporous carbon (NPC) and polydimethylsiloxane (PDMS) can be used. For example, these films were demonstrated to prepare centimeter-scale membranes in our previous work (*Nature Energy* 9, no. 8 (2024): 964-974). We have now added following text to the manuscript:

“Membranes exhibited a modest decline in CO₂ permeance (~15%) over 60 hours of testing. However, a simple thermal treatment at 130 °C for 2 h enabled partial performance recovery (Figure S15).”

Figure S15. 70 h performance stability test of porous graphene membrane.

REVIEWER 1, ATTACHMENT 1

Comments:

I think the idea is good, and work is excellent. I would like to recommend the publication of this paper after the following few comments are addressed.

1. I cannot find the novelty of this work. Similar type of work has been already published by same authors (<https://www.nature.com/articles/s44286-025-00203-z#Sec15>)
2. It is suggested that authors shall added key results in abstract to reflect the worth of manuscript.
3. A minor comment on terminology, see J. Membr. Sci. 2010, 348, 346-352. Nature Communications, 2018, 9, 486., Flux should be in a unit of $\text{Lm}^{-2}\text{h}^{-1}$, permeance should be in a unit of $\text{Lm}^{-2}\text{h}^{-1}\text{bar}^{-1}$, permeability should be the permeance normalized with thickness. Usually flux is used for organic separation, etc. Therefore, use appropriate terminology in MS.
4. Please check the “High-temperature oxidation (e.g., at 80 °C).....
5. It is recommended to improve introduction section. Relevant queries / questions of research should be addressed in the introduction and try to avoid the AI use.
6. Figure 3b-d showed some contamination. Can authors explain the reason for it and how we can avoid it? Is there effect of these contaminations on results?
7. Authors shall also focus on XRD of materials in revised work
8. Authors explained that “Even 5 min of post-ozone exposure nearly doubled the ID/IG ratio from 1.10 to 2.25 (Figure 5b). Longer treatments (15–30 min) further broadened the D, G, and 2D bands, diminished 2D peak intensity, and produced a pronounced (D + D') band, signifying greater lattice disorder. Did authors checked further longer treatment?
9. Did authors compared their results with literature?